# Sensing the DNA-mismatch tolerance of catalytically inactive Cas9 via barcoded DNA nanostructures in solid-state nanopores

Sarah E. Sandler [1], Nicole E. Weckman [1,4], Sarah Yorke [1,5], Akashaditya Das[2,6], Kaikai Chen [1], Richard Gutierrez[3] & Ulrich F. Keyser[1] ✉

Single-molecule quantification of the strength and sequence specificity of interactions between proteins and nucleic acids would facilitate the probing of protein–DNA binding. Here we show that binding events between the catalytically inactive Cas9 ribonucleoprotein and any pre-defined short sequence of double-stranded DNA can be identified by sensing changes in ionic current as suitably designed barcoded linear DNA nanostructures with Cas9-binding double-stranded DNA overhangs translocate through solid-state nanopores. We designed barcoded DNA nanostructures to study the relationships between DNA sequence and the DNA-binding specificity, DNA-binding efficiency and DNA-mismatch tolerance of Cas9 at the single-nucleotide level. Nanopore-based sensing of DNA-barcoded nanostructures may help to improve the design of efficient and specific ribonucleoproteins for biomedical applications, and could be developed into sensitive protein-sensing assays.

The recognition of nucleic acid sequences by proteins is fundamental to biology. Owing to the specificity of protein–DNA binding, interactions between these biomolecules form the basis of many biosensing tools, biomolecular engineering techniques and new therapies[1,2]. In particular, clustered regularly interspaced short palindromic repeats (CRISPR)/dCas9 systems have emerged as powerful tools for targeted gene expression as well as diagnostics, and substantial work is currently being undertaken to improve the efficiency and specificity of CRISPR/Cas[3,4]. It is therefore essential to develop simple and sensitive assays that analyse the interactions between nucleic acids and proteins in their folded and functional states, to capture their native behaviour. Furthermore, it is imperative that these methodologies be sensitive to single-nucleotide differences in sequence, which can dramatically impact binding.

Resistive pulse sensing with solid-state nanopores is an appealing method to assess binding events. The technique offers the flexibility to study DNA, RNA and proteins in their native states with a single sensing system. Nanopore sensing relies on measuring changes in ionic current as molecules are driven through a nanometric pore using an applied electric field. The ionic-current change owing to a translocation event reflects the size, shape and charge of the molecule[5]. There are two types of nanopore: biological and solid-state. Although biologically derived protein nanopores are ideally suited for DNA sequencing[6], solid-state nanopores offer control over pore size during the fabrication process, allowing for the detection of a wide range of analytes.

Although this versatility permits the analysis of a variety of biomolecules, the lack of specificity in sensing also presents a challenge for multiplexed detection, wherein the targets of interest should be easily differentiated. To overcome this barrier, we capitalize on DNA nanotechnology, which enables the design and assembly of custom nanostructures that contain specific binding sites for proteins of interest[7].

In this Article, we fabricate solid-state nanopores by pulling quartz capillaries, also known as nanopipettes, to be used as a general

[1]Cavendish Laboratory, University of Cambridge, Cambridge, UK. [2]Department of Pathology, University of Cambridge, Cambridge, UK. [3]Oxford Nanopore Technologies, Oxford Science Park, Oxford, UK. [4]Present address: Institute for Studies in Transdisciplinary Engineering Education & Practice, Department of Chemical Engineering & Applied Chemistry, University of Toronto, Toronto, Canada. [5]Present address: Yusuf Hamied Department of Chemistry, Cambridge, UK. [6]Present address: Department of Chemical Engineering, Imperial College London, London, UK. ✉e-mail: ufk20@cam.ac.uk

detection tool. Double-stranded DNA (dsDNA) creates an ionic current drop in the nanopore signal. When binding an analyte such as a protein to the dsDNA, secondary current spikes are created. By designing specific binding sites or sequences along a DNA strand, an identifiable pattern is produced. This fingerprint is observable as a unique signature of spikes in the ionic current. The labelling and mapping of targeted native DNA sequences in nanopores has previously been accomplished using a variety of methods, including peptide nucleic acid probes[8,9], biochemical ligation[10], transcription factors[11] and chemical modification with methyltransferase and biotinylation[12]. Recently, the use of nanopore analysis to measure the binding of a variant of catalytically inactive or dead Cas9 (dCas9) to dsDNA has been demonstrated[2,13] Our system builds upon these initial proof-of-concept works by introducing designed DNA nanostructures for user-defined DNA target testing, thus expanding the application space of nanopore sensing, DNA nanotechnology and screening of protein–DNA interactions. Here we demonstrate that, when carefully designed and tested, dCas9 complexes could act as a label allowing highly specific mapping of the DNA to determine single base-pair changes, which is particularly important for diagnostics[4,14].

The dCas9 protein is a catalytically inactivated form of the CRISPR-associated protein, Cas9, that will bind, but not cut, a 20-base-pair (bp) target dsDNA sequence. The dCas9 ribonucleoprotein (RNP) is a complex assembled with a dCas9 protein and guide RNA (gRNA). The gRNA consists of two RNA strands including the CRISPR RNA (crRNA) which is complementary to the target dsDNA sequence, as well as a trans-activating crRNA (tracrRNA). The binding site of the dCas9 to the dsDNA is programmed by changing the sequence of the crRNA. The RNP binds in the presence of an NGG protospacer adjacent motif (PAM) and will then hybridize to 8–12 target nucleotides upstream of the PAM, also known as the 'seed' region. Mismatches in the seed region will prevent the RNP forming a complex with the target, whereas mismatches downstream in the 'distal' region primarily act to reduce the lifetime of the complexes[15].

By targeting the dCas9 to different dsDNA sequences, unique and sequence specific patterns or 'barcodes' of bound dCas9 can be created and used for multiplexed identification of DNA using a nanopore[2]. To capture the full potential of these dCas9-based systems, however, they must be quantitative, scalable and specific, demanding the screening of dCas9 to select for high binding efficiency and specificity. Binding is highly dependent on the gRNA sequence; while in some cases this interaction is incredibly specific, other sequences may possess hundreds of off-target sites, limiting the utility of these systems[16]. Current methods for assessing the binding of dCas9 to DNA are primarily based on sequencing and computational modelling[17–20].

Computational methods are very useful to guide experimental design, but are limited due to their sensitivity to data fluctuation and insufficient training data[21]. Moreover, although mismatches are known to impact the probability and stability of binding events for dCas9, it is difficult to create a single set of general rules for assessing mismatch tolerance for a number of reasons[21]. Firstly, almost all computational tools available are based on large-scale experimental datasets geared towards predicting cleavage, not binding, and it has been shown for Cas9 the relationship between binding and cleavage is complex[18,22]. An additional layer of complexity arises from evidence that cleavage efficiency is largely dependent on gRNA sequence, with each probe having unique guide-intrinsic mismatch tolerance (GMT)[23,24]. GMT makes it difficult to predict the specificity of a probe without testing it, highlighting the need for a tool to rapidly screen assay probes.

By pairing single-molecule nanopore measurements with DNA nanotechnology, we have created a system that can uniquely and accurately assess the specific binding interactions of dCas9 probes and target dsDNA. Through the design of DNA nanostructures patterned with engineered dumbbell barcodes at one end of the nanostructure, we created a unique spike pattern that corresponds to a specific target

sequence for the dCas9 (ref. 7). We expanded the utility by adding designed dsDNA overhangs, which allowed us to rapidly assess the sequence-specific gRNA binding specificity of dCas9 probes to any short DNA sequence of interest. In addition to measuring dCas9 binding to the matching target sequence, the dsDNA overhang can be designed with mutations to test the sensitivity of the crRNA to these modifications. We show that these methods are quantitative, and that they can differentiate varying concentrations of DNA with and without introduced mutations in the same sample. This approach of testing gRNAs may facilitate the design of dCas9 RNPs used for gene-expression and diagnostic applications to be more efficient and specific.

## Results

### Evaluating sequence-specific dCas9 binding in nanopores

DNA nanostructures have previously been shown to be useful tools for the multiplexed measurement of biomolecule binding using the nanopore sensing system[7]. Figure 1 introduces the design of the DNA nanostructure and nanopore sensing system and demonstrates its use for detecting the binding of a dCas9 RNP to a particular target dsDNA overhang sequence.

In Fig. 1a we show a schematic of the DNA nanostructure. As described in detail in Methods, the DNA nanostructures are created from a single-stranded DNA (ssDNA) backbone, with staple oligos complementary to the backbone that bind along its length. The sections of the nanostructure that are fully complementary to the ssDNA backbone strand appear in a translocation event as a current drop corresponding to dsDNA. A section of the staple oligos is replaced, and a portion of the DNA nanostructure is patterned into a DNA barcode consisting of five spikes which can be set to either '0' or '1'. The barcode design is based on previous work that was optimized to create clearly distinguishable spikes in nanopores with diameters around 15 nm (ref. 7). While this barcode design leaves us with only $2^5$ (32) sequences that can be tested, we have already explored the maximum limits of barcodes. We can increase the density of the nanostructures and the signal-to-noise ratio using smaller nanopores with 56 bits on a single DNA carrier, allowing for a library of $2^{56}$ (>$10^{16}$) molecules[25]. The number of multiplexed protein–DNA interactions is limited only by the number of nanopores and our ability to assemble these DNA nanostructures. In the sensing region of the nanostructure, two oligos were designed to create a dsDNA overhang that is 50 bp in length. This dsDNA overhang can present any DNA sequence of interest including target sequences for dCas9 binding.

The different nanostructures depicted in Fig. 1b can be combined in solution and translocate through the nanopore one by one. Two different barcoded nanostructures 11111 and 11001 are shown in this schematic with bound dCas9 and unbound dCas9 along with the signals generated by the structures during nanopore translocation (Fig. 1c). A raw current trace both with and without DNA/protein can be seen in Supplementary Fig. 1. Translocation can be in either direction as seen in Supplementary Fig. 2. When dCas9 does not bind to the target dsDNA in the overhang region, there is the absence of the additional spike. The overhang region contains only 50 bp of dsDNA below the detection limit of our nanopores and hence generates no clear spike in the signal. Thus, the presence or absence of dCas9 binding to the nanostructure can be clearly distinguished by a simple thresholding algorithm around the expected spike location.

### Quantifying DNA nanostructures using dCas9 binding

The single base specificity of dCas9 gives it an advantage as a diagnostic tool to probe point mutations in clinically relevant sequences. The single point mutation in the *katG*315 gene in *Mycobacterium tuberculosis* is responsible for antibiotic resistance to isoniazid for 64.2% of resistant cases[26]. The target DNA region that contains this mutation was selected as seen in Fig. 2a. Before using dCas9 RNPs as a diagnostic tool, creating a method to test the specificity of the probe is essential. This was tested

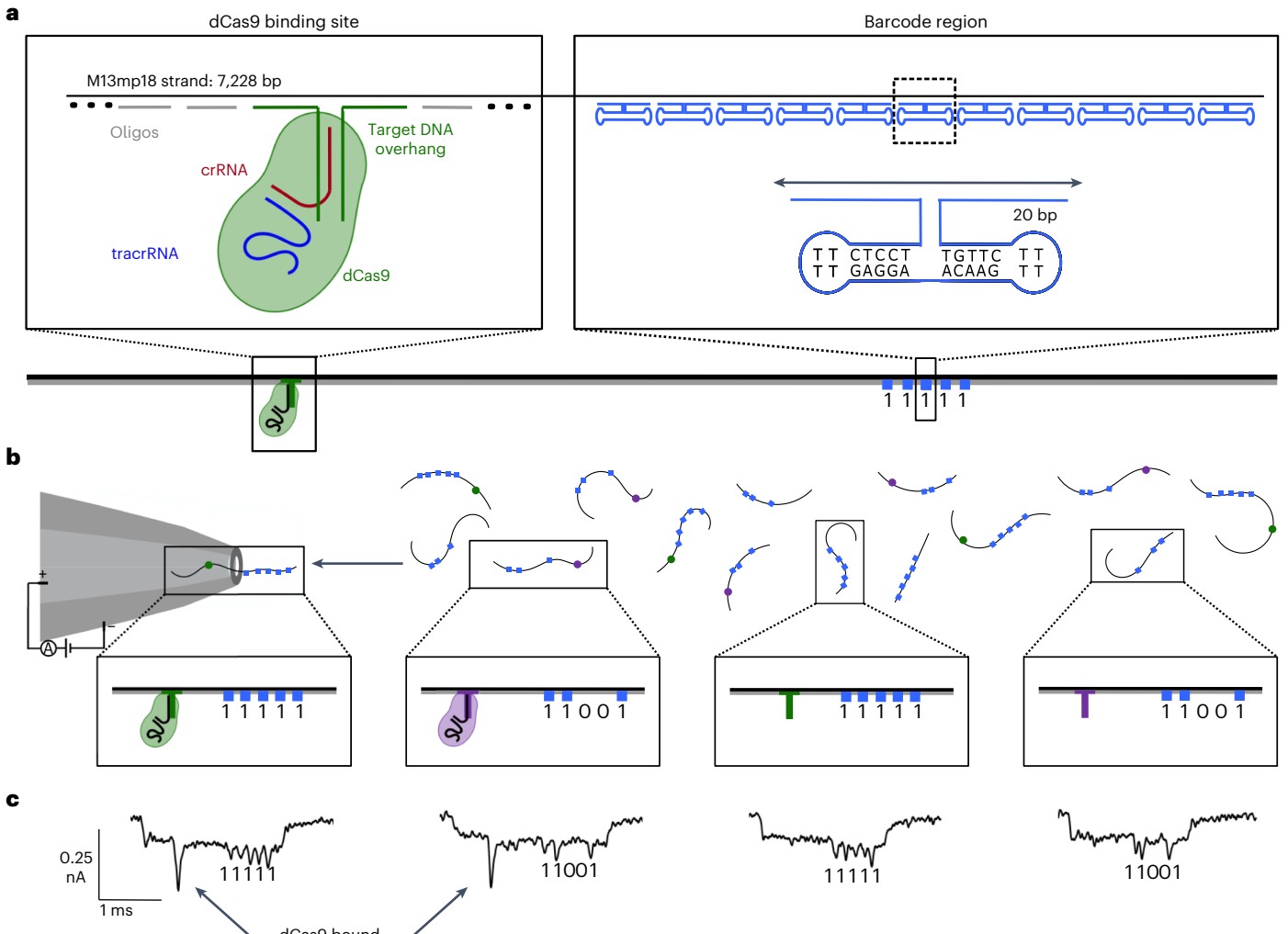

**Fig. 1 | DNA nanostructures for assessing the specific binding dCas9 to target sequences. a**, Schematic of DNA structure with DNA barcode region and dCas9 overhang. Both the sequences used for the dumbbell region that creates the barcode (blue) and the schematic of the target DNA region that acts as an overhang (green) are highlighted. Additionally, the components involved in the dCas9 RNP are shown. **b**, Solid-state nanopore with two different DNA nanostructures (11111 and 11001) mixed together in solution. Both these structures are shown with and without dCas9 binding, highlighting the capability to determine specificity and binding efficiency. **c**, Current traces from the nanopore of the different DNA nanostructures when two probes are added. The 50 bp overhang cannot be resolved unless the dCas9 is bound.

by creating two different nanostructures with different barcodes and different overhangs with DNA target regions identical to the wild-type genome and the resistant genome with the mutation.

We added one of the dCas9 probes at a time to equimolar concentrations of the two DNA nanostructures as seen in Fig. 2a. This was done to test the binding efficiency and specificity of each probe in the presence of the single base-pair change as seen in Fig. 2b. We found that both of the dCas9 probes had high specificity to their perfectly matched target overhang with over 94.7% and 94.0%, respectively, of the DNA nanostructures events with clear barcodes being correctly labelled. The nanostructures with a mutation that mismatched to the probe being tested were found to have binding of 5.3% and 6% respectively. Similarly, a control done with no PAM sequence and the purple probe was found to have a binding of 4.2% ($N = 169$). These false positives are most likely due to knots in the DNA during translocation which may produce similar signals to a bound dCas9 protein in the event trace. The equations used to calculate these values can be seen in Methods. The number of events differs between experiments because of the percentage of clearly labelled unfolded events that are used in data analysis after measurement. Our results show that, with probes

that are carefully designed and tested, we can differentiate binding to dsDNA targets with single nucleotide specificity.

Another crucial capability for biosensing tools, is the ability to quantify relative abundance of target sequences. We tested the quantitative nature of the binding of dCas9 probes to their target DNA by mixing equimolar amounts of two crRNA probes and varying the ratio of the two different target DNA nanostructures identifiable by their respective barcodes, as seen in Fig. 2c. Using relative concentrations of DNA nanostructures with each barcode in ratios 0:100, 25:75, 50:50, 75:25 and 100:0, the ratio of events with the barcode and signature dCas9 spike was found to be consistent with the relative concentration added as can be seen in Fig. 2c. For example, in these experiments, before dilution for nanopore experiments, a total of 3 nM of DNA is mixed with 50 nM of each formed RNP. Thus, if one adds 0.75 nM of 10011 barcode nanostructure and 2.25 nM of 11111 barcode nanostructure, upon adding equimolar concentrations of the RNPs one will still find the percentage of nanostructure events with the 10011 barcode and a dCas9 spike will be close to 25% and the number of nanostructure events with a 11111 barcode and a dCas9 spike will be close to 75%. If one knows the concentration of 10011 added was 0.75 nM, they can

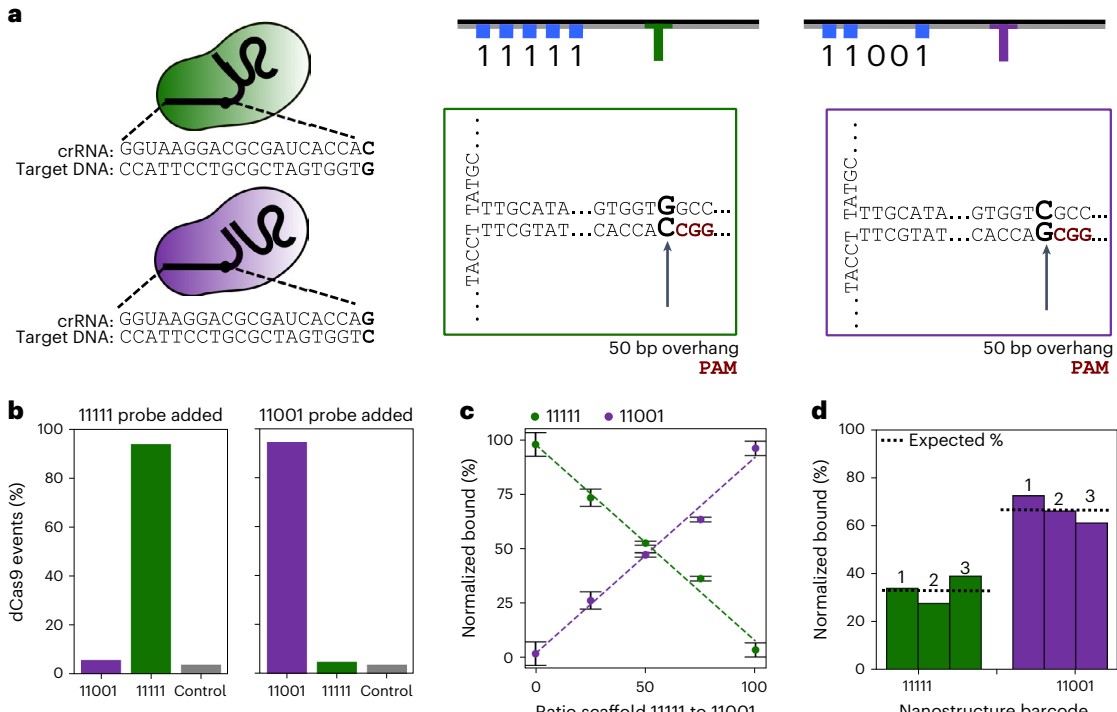

**Fig. 2 | Quantifying dCas9 binding to the nanostructures. a**, dCas9 probes and DNA nanostructures overhang schematics depicting single base difference between sequences in crRNA probe (with the PAM in red) and DNA overhang. The single base-pair change can be found in Kat315 gene in isoniazid-resistant *Mycobacterium tuberculosis*. Two different DNA nanostructures with different overhangs (green and purple) are depicted. **b**, Binding specificity of the probes when both DNA nanostructures are added and only one respective probe is added. $N > 65$ events for each experiment. A control nanostructure with target DNA containing no PAM was also tested as seen in grey, where $N > 165$. Calculations were made using equations (1) and (2). **c**, Quantification of labelled events dependent on varying relative concentration ratio of DNA nanostructures (green or purple) in solution with equimolar concentrations of the two dCas9 probes (standard deviation represents one sample measured in at least two different pores). Calculations were made using equations (3–6). $N > 85$ events for each condition. Error bars represent standard deviation between nanopores. **d**, Reproduced experiments with 11001 and 11111 nanostructures barcode at a ratio of 2:1 in solution with equimolar concentrations of both dCas9 probes added. The different bars (1, 2 and 3) are three independent sample preparation repeats in three different nanopores. Calculations were made using equations (3–6). $N > 100$ events for each measurement. In all experiments, dCas9 probes were added in excess to target DNA molecules.

infer through relative concentration that the other barcode would be 2.25 nM. These percentages were normalized to the measured binding efficiencies for each probe as seen in Methods. We measured the samples in at least three different pores to account for variability in pore fabrication, as shown by the error bars, with consistent results independent of the pore used. The reproducibility was further tested by testing three independent repeats from the step of sample preparation in three different pores when mixing the structures in a 2:1 ratio. The standard deviation between the independent repeats was only ~6% as seen in Fig. 2d. Furthermore, by adding several dCas9 probes to a mixed DNA sample, our method can be used to quantify the relative concentrations of different DNA targets in solution. This combination of quantification of relative DNA concentrations in a mixture as well as the specificity for detecting single base-pair changes is a proof of concept for a powerful nanopore sensor for multiplexed DNA analysis.

The ability to use CRISPR/dCas9 probes to quantitatively measure the amount of a given DNA sequence present in a sample has many implications. The concentration of DNA in samples has been measured in nanopores by methods such as the electrophoretic capture model[27] and controlled counting method[28,29]. However, methods, such as the controlled counting method, rely on a DNA control that has a notably different length than the unknown sample[30]. While our method requires an internal standard for quantitative detection, there is no need for pre-existing knowledge of the size of the target DNA. The specificity of the method is determined by the bound dCas9 probes. One can use DNA markers or several dCas9 probes to create a pre-designed

spike pattern on the DNA. Spike patterns identify a DNA marker at a known concentration by mixing with a sample of DNA at an unknown concentration. The combination allows one to determine the relative concentration of the target.

## Single nucleotide specificity on varying position and identity
Understanding the target specificity of probes based on both position and base-pair change of the mutation is essential for designing dCas9 probes both for genomic engineering and for CRISPR diagnostic techniques. This was studied using a different DNA nanostructure with three overhangs highlighted in Fig. 3a,b. As seen in Fig. 3b, we introduced single base-pair changes in the overhangs at both the PAM site and at various positions from the PAM site to compare the effect both position and base identity mismatch. The sequences for these overhangs can be found in Supplementary Table 4 and the sequences for dumbbells that generate spikes can be seen in Supplementary Table 5. The binding of a single dCas9 RNP complex (Fig. 3c) was tested against the different mismatched DNA nanostructures in the nanopore. Example events can be seen in Fig. 3d. The normalized binding ratio was then calculated for the probe at the different positions with the different base-pair identities and plotted in Fig. 3e. The equations used to calculate normalized binding ratio can be found in Methods. These calculations rely on normalizing to the value for binding to the control ($X_{control}$ (10)), which was 33.7% with a 2.4% standard deviation as seen in Supplementary Fig. 3. Thus, if a probe is found to have a binding percentage of 33.7%, it would correspond to a normalized binding

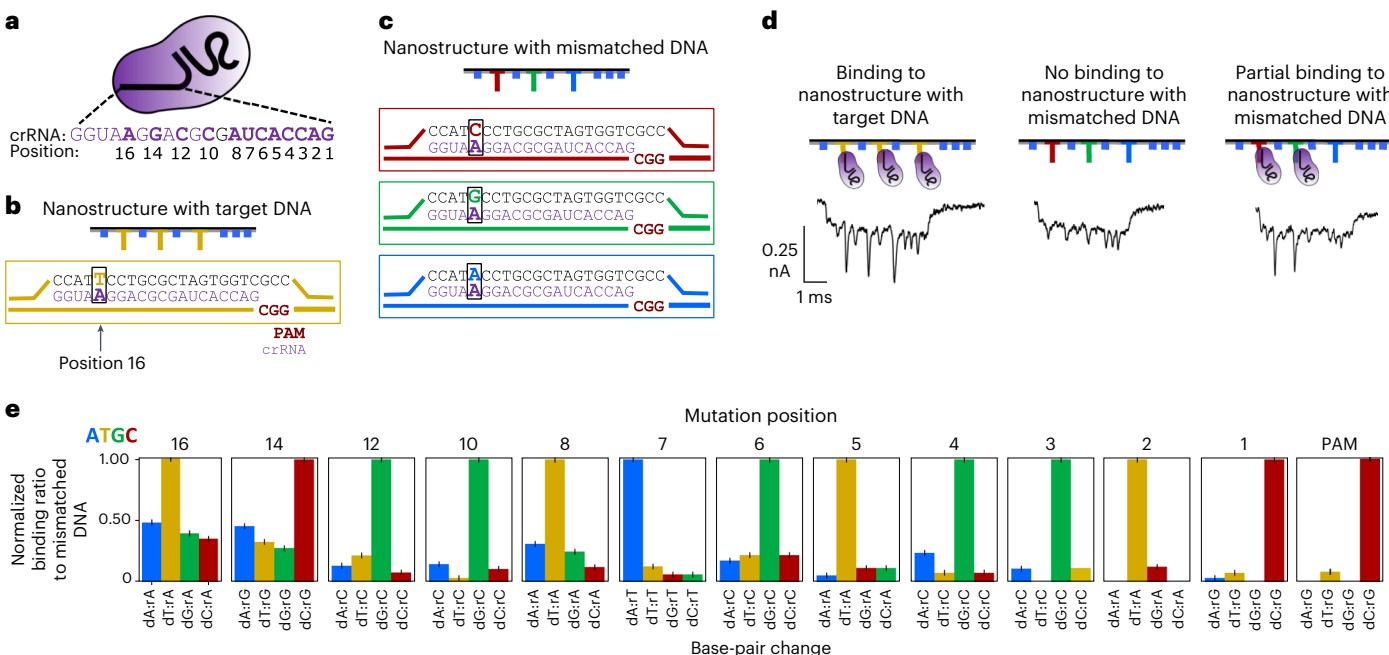

**Fig. 3 | Measuring sensitivity to single base-pair changes. a**, The dCas9 RNP used for assessing specificity to mutations. **b**, Control nanostructure with target sequence without any mutations in overhang region. **c**, Nanostructure with overhangs with mismatches in the target DNA overhang region. **d**, Schematic of nanostructure showing dCas9 binding to overhang and corresponding nanopore event trace. The furthest on the left represents binding of the RNP to a nanostructure with three overhangs that are control. The middle event represents the three mismatched overhangs with no observed binding of the RNP. The furthest of the right represents the dCas9 RNP binding to two of the mismatched target sequences, highlighting low specificity. **e**, Bar plots for each possible different DNA base mutation and position with normalized binding ratio being calculated using equations (7–11). *N* > 45 events for each experiment. Error bars are result of standard deviation from normalization to control binding efficiency.

ratio of 1.0. The normalization is an essential step because this lower binding efficiency for these experiments is due to an increase in salt concentration in the buffer. With more DNA dumbbells, a higher salt concentration must be used to slow translocation for the nanopore, as seen in Supplementary Fig. 4; however, this does reduce efficiency of binding, as seen in Supplementary Fig. 5.

Mutations one, two and three base pairs proximal to the PAM show the lowest binding efficiencies, suggesting that this site has the highest specificity for dCas9 binding for this given probe. As the mutations near the PAM distal region, there is a steady increase in dCas9 binding. This is consistent with findings in the literature that PAM proximal bases are more crucial for successful identification of the PAM site and the subsequent binding by the dCas9 (ref. 31). The findings also agree with literature in that bases in the PAM distal region of dCas9 are not as specific for dCas9 binding[31].

Consistent with the literature, the DNA with mismatches with the highest relative binding occur due to the mispairs adopting 'wobble' base pairing or Watson–Crick-like conformers[32–34]. The most tolerated base pairing (rG-dT, rU-dG, rA-dC and rC-dA) is a result of the flexible DNA backbone enabling slight shifts in the position of nucleotides, hence the name 'wobble'. Our investigation of the most prevalent mismatched DNA–RNA base pairings shows these wobble base pairs to be dominant from the PAM site to site 5. This suggests that, near the PAM site, mismatches in the base pairing are tolerated due to the flexible DNA backbone misaligning nucleotides, and that these mismatches do not hinder successful DNA–RNA base pairing. However, bases further from the PAM site, starting at position 5, do not show this preference for wobble mismatch. This perhaps suggests that, as the mutations travel further from the key PAM 'anchoring' site, there are other external factors contributing to the mismatched base pairing. The theory that the 'wobble' base pairing influences specificity agrees with recent works where the incorporation of inosine into dCas9 gRNAs reduces binding

to cognate DNA sequences, but allows for pairing with sequences bearing single substitutions at overlapping position[35]. Additionally, the structural basis of wobble effects in the context of Cas9 has been established in the literature, supporting these claims[36].

Evaluating single-mismatch specificity has been highly investigated in the literature with a number of different probes. While consistently most probes show the PAM proximal/distal trend and the wobble-base trend, each guide differs from paper to paper. For example, in our study, we found a decrease in specificity, relative to the general trend, at position 8, and a notably increased specificity at position 10. Previous research, although measuring cleavage efficiency, has shown that the core sequence sensitive to mismatch is at base pairs 4–7 upstream of the PAM[37]. Other research, however, has shown no major difference between positions 2 versus 4 for cleavage efficiency, but has demonstrated there is an increase in sensitivity to mismatches at positions 11 and 13 (ref. 38). These observations are consistent with the concept that probes possess varying degrees of GMT, which we go on to investigate in Fig. 4.

### Effect of guide-intrinsic mismatch on binding

To highlight the usefulness of this assay tool, two probes, depicted in Fig. 4a, were compared. These probes were selected because of their varying prediction scores using CHOPCHOP but similar binding efficiencies to target DNA structures as seen in Fig. 4b (ref. 39). Despite the prediction efficiency score for probe 2 being low, it was found to have a high binding efficiency. This gap between predicted and measured may be due to the predicted score being based on cleavage efficiency rather than binding, as it is known the two are not directly interchangeable[22]. However, because of the lack of computation tools to predict binding, CHOPCHOP, which provides a predicted cleavage efficiency value was used. Because of the similar binding efficiency, it was postulated the probes may have similar trends in single base-pair

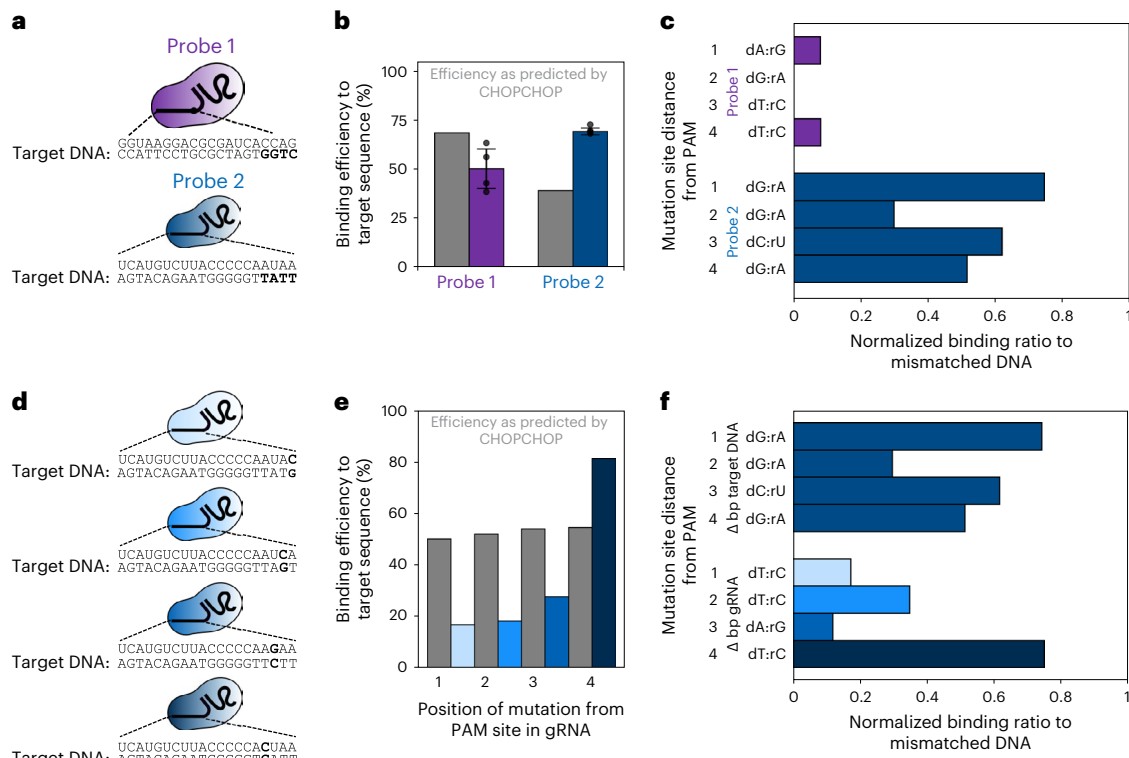

**Fig. 4 | Probe-to-probe variation on the specificity of the Cas9 complex.**
**a**, Probe 1 and probe 2 were designed and compared using nanostructures described in Fig. 1. **b**, Comparing binding efficiency of probes to each other and to the predicted efficiency from CHOPCHOP (grey)[39]. $N = 288$ events for probe 1 and $N = 558$ events for probe 2. Error bars are the standard deviation between at least four different measurements in different pores for each probe. **c**, Bar graphs of binding ratio normalized to efficiency of probes to target. Sensitivity to single base-pair changes varies greatly between the two probes, highlighting influence of GMT. Each experiment has at least $N > 45$ events. **d**, Variations of probe 2 with mutations in gRNA. **e**, Comparison between predicted efficiency (grey) and measured binding efficiency when mutations are introduced into gRNA. Each experiment has at least greater than $N > 55$ events. **f**, Bar graphs with normalized binding ratios depicting probe dependence of GMT and large variation of binding among probes with introduced mutations. Each experiment has at least more than $N > 55$ events.

specificity. The first four base pairs from the PAM were interchanged with bases that were not wobble base pairs, which would suggest the probes should have a higher level of specificity. However, probe 2 showed to have very little observed sensitivity to single base-pair mutations with a very high normalized binding efficiency as seen in Fig. 4c. Despite the high binding efficiency, this lack of specificity highlights that this probe would not be useful as a tool for detecting single base-pair changes in clinical samples. In contrast, probe 1 shown in Fig. 4c is a highly specific probe, which would make it useful as a diagnostic tool. This demonstrates the need for an assay such as the one demonstrated this work to test specificity of dCas9 probes before implementation in further experiments. Because of this interesting finding, this probe was then tested further by introducing single base-pair changes into the gRNA sequence as seen in Fig. 4d. Despite similar predicted binding efficiencies between the four probes, as seen in Fig. 4e, the normalized binding efficiencies that were measured varied considerably. This highlights the claims in the literature that the efficiency of the dCas9 and Cas9 RNP depends uniquely on the gRNA being used, with different guides having different GMTs[23,24]. Introducing single base-pair mismatches in the DNA has a completely different effect to introducing base-pair changes in the RNA at the same location as seen in Fig. 4f. Just one single base-pair change among guides can result in completely different behaviour and sensitivity to single base-pair changes. Because of these behaviours that are unique to guide and the current limitations of computational tools to predict these behaviours, the assay technique as demonstrated in this work is essential to informed dCas9 probe design.

## Discussion

In this work, we highlight the potential of using dCas9 probes for the highly specific detection of base-pair changes in DNA. We differentiated mixed sequences with single base-pair specificity, and quantified the relative concentrations of DNA using dCas9 probes. However, to use dCas9 probes for these applications, it is essential to first evaluate the specificity of the probe. Hence, we have presented a model assay that is sensitive and specific for testing dCas9 probes. This assay can be used to verify the specificity of probes before they are used in diagnostic and gene-editing contexts.

Although currently this system is limited to the detection of dCas9 binding, there is potential for translation as a tool to measure cleavage as well. It has been shown that SpCas9 remains bound very stably to DNA after cleavage[40]. Similarly, Cas12a cleaves the PAM distal region of the target, but remains bound in the PAM proximal[41]. There is potential to innovate DNA nanostructure designs similar to the one discussed in this work and test how the cleavage of a variety of different enzymes can be affected by mismatches in the target DNA.

Beyond testing the specificity, we illustrate the utility of these probes in the diagnostic realm as a method of determining relative concentration. Concentration measurements are particularly important in diseases or illnesses. The relative abundance of a certain bacteria or virus is relevant to determine the treatment, or to justify a change in treatment for example due to increasing concentrations of a resistance marker. As we have shown here, an additional benefit is that dCas9 probes is that they can potentially be used to detect relative concentrations of a pathogen with a specific mutation.

Our methodology has the potential to be translated to applications beyond those shown in this work in both diagnostics and gene editing. With an expanded set of barcodes, it can be used as a highly specific, high-throughput approach to assaying the dCas9 RNPs to test hundreds of gRNAs in the same measurement. While there are in vitro and cell-based high-throughput techniques, such as chromatin immunoprecipitation followed by sequencing[42], it may be excessive depending on the final application. One advantage of the nanopore system is a few sequences of interest can be tested with minimal sample amounts (fM concentrations per experiment), which can save the user time and resources. A reduction of enzyme amounts as well as target samples is especially relevant in diagnostic use cases[1,43] and in investigations of dCas9 inhibition[44], where the electrophoresis mobility-shift assay is the de facto standard. In terms of developing diagnostics, using dCas9 as a label on native DNA has already been demonstrated[2], and by using the nanopore DNA-nanostructure system one can efficiently design and test dCas9 probes.

The combination of nanopore sensing and CRISPR/Cas molecular engineering techniques enables a single-molecule highly specific approach to assessing the binding efficiency and specificity of CRISPR/dCas9 probes. To this end, we have created DNA nanostructures containing a DNA overhang that can be designed to have different dCas9 binding sites, with different 'barcodes' identifying the overhang sequence to enable multiplexed measurements. We have shown that this system can be used for assessing the specificity of dCas9 probes and the guide-intrinsic mismatch intolerance of different probes. This is particularly important when using these probes in diagnostic applications for the detection of single base-pair changes. Compared with traditional techniques, assessing binding with this system provides advantages in speed and specificity.

## Methods

### DNA-nanostructure barcodes

To observe that the CRISPR–dCas9 system is specific enough to detect the single base pair, DNA constructs with different 'barcodes' plus an overhang sequence for dCas9 was created. The DNA construct was synthesized from pairing a linearized 7.2 kbp single-stranded (ss) M13mp18 DNA (GuildBiosciences) with 190 complementary oligonucleotides via Watson–Crick base pairing to produce full dsDNA over the period of 1 h in a thermocycler. All oligonucleotides were synthesized by Integrated DNA Technologies and dissolved in IDTE (10 mM Tris–HCl and 0.1 mM ethylenediaminetetraacetic acid, pH 8.0), and the sequences can be found in Supplementary Information. The sample is then filtered using a 100 kDa Amicon filter and measured in a nanodrop spectrophotometer for concentration information. Based on the nanodrop measurement, typical yield is 75–95%. For the nanostructures in Fig. 1, within the 190 oligos are five groups of equally spaced simple dumbbell hairpin motifs to create the spikes that act as a barcode on the DNA nanostructure[7]. Each group consists of 11 DNA dumbbells to create a single spike. The exact sequences with their numbers are shown in Supplementary Table 1 in Supplementary Information following a previous work[7]. The overhang was created by replacing oligos nos. 142 and 143 with a 90 bp oligo made up of 30 bp segments to match the M13 backbone and 50 bp of the specific sequences containing the target sequences we aimed to test. The 50 bp dsDNA overhang is not large enough to generate a current blockade that can be observed. These overhangs are provided in Supplementary Table 2 for the experiments in Fig. 2 and Supplementary Table 3 for the experiments in Fig. 4. For the second nanostructure, shown in Supplementary Fig. 3, oligos no. 44, 45, 81, 82, 118 and 119 were replaced with overhang sequences as found in Supplementary Table 4. The dumbbells were made by replacing oligos no. 23–28, 60–65, 97–102, 134–139, 148–153 and 162–167 with the sequences in Supplementary Table 5. All samples were stored in a storage buffer of 10 mM Tris 0.5 mM MgCl$_2$.

### Design of dCas9 probes and binding

Catalytically deactivated Cas9 D10A/H10A (dCas9) from *Streptococcus pyogenes* binds with a tracrRNA and a sequence-specific RNA (crRNA), both synthesized by Integrated DNA Technologies and dissolved in IDTE (10 mM Tris–HCl and 0.1 mM ethylenediaminetetraacetic acid, pH 8.0). The target sequences for the crRNA for the probes were designed using online software (http://chopchop.cbu.uib.no/)[39] and can be found in Supplementary Information. To assemble the dCas9 RNPs, the tracrRNA (200 nM), crRNA (250 nM) and dCas9 (100 nM) were incubated in a low-salt buffer (25 mM HEPES–NaOH (pH 8.0), 150 mM NaCl and 1 mM MgCl$_2$) at 25 °C for at least 20 min.

The assembled dCas9 probes were then incubated with the DNA nanostructures for at least 20 min at 25 °C, with the dCas9 probes added in excess of typically 15 dCas9 probes per DNA binding site. The samples containing DNA nanostructures labelled with dCas9 are diluted to 0.1–0.3 nM into a 2 M LiCl, 1× TE buffer solution or 4 M LiCl, 2× TE, depending upon the nanostructure, immediately before the beginning of the measurement in the nanopore system.

### Nanopore fabrication and measurement

Nanopores are fabricated from commercially available quartz capillaries (0.2 mm inner diameter/0.5 mm outer diameter Sutter Instruments) using a laser-assisted pipette puller (P-2000, Sutter Instrument) to around 15 nanometres. We produced a polydimethylsiloxane (PDMS) chip with 16 conical nanopores with a communal *cis* reservoir and individual *trans* reservoirs. Detailed instructions for production can be found at Bell et al.[45]. Silver/silver-chloride (Ag/AgCl) electrodes are connected to the *cis* and *trans* reservoirs in the polydimethylsiloxane chip. The size of each nanopore is estimated before beginning measurements by taking a current–voltage curve in the baseline electrolyte. The central *cis* reservoir contains the sample and is grounded, while a 500 mV bias voltage is applied to the *trans* reservoir to drive DNA transport through the nanopore. The measurement is then taken until around 1,000 folded and unfolded events are gathered, with a typical time range of 45 min to 2 h depending upon the concentration used and nanopore. Typically, of these 1,000 events, 300 are unfolded and then analysed. An example of the folded and unfolded events can be seen in Supplementary Fig. 6.

The Axopatch 200B patch-clamp amplifier (Molecular Devices) was used to collect current signals. The set-up is operated in whole-cell mode with the internal filter set to 100 kHz. To reduce noise, an eight-pole analogue low-pass Bessel filter (900CT, Frequency Devices) with a cut-off frequency of 50 kHz is also used. The applied voltage is controlled through an I/O analogue-to-digital converter (DAQ-cards, PCIe-6251, National Instruments), using a program on LabView 2016 to simultaneously record the current signal at a bandwidth of 250 kHz.

### Analysis of nanopore data

From the Labview GUI, experimental data are stored as technical data management streaming (TDMS) files. First, a translocation finder Python script (part of the nanopyre package found at https://gitlab.com/keyserlab/nanopyre) is used that identifies the events from the raw traces and stores them in an hdf5 file. After the initial translocation finder analysis, the events from the hdf5 files are read into Python (using the nanopro package https://gitlab.com/keyserlab/nanopro) and all events with current noise >15 pA are discarded. The parameters to find the spikes are based on manual analysis of the threshold, height, distance and prominence parameters from the Python peakfinder package. This is tested on the first ten and last ten events to ensure the parameters are consistent and will accurately find the peaks. Following this, events are sorted on the basis of the number of spikes. Following the sorting, the events are analysed by eye and events that have folds or knots interfering with the barcode are discarded. Our lab has shown that as few as four events are sufficient for positive detection in the majority of cases, while nine correct events increase the probability

of positive detection to more than 90% (ref. 46). Percentage of events with dCas9 bound in Figs. 2b and 4 is calculated the following way:

$$\%\text{dCas9 Events}_{11111} = \frac{N_{11111\,dCas9}}{N_{11111\,dCas9} + N_{11001\,dCas9}} \times 100 \qquad (1)$$

$$\%\text{dCas9 Events}_{11001} = \frac{N_{11001\,dCas9}}{N_{11111\,dCas9} + N_{11001\,dCas9}} \times 100 \qquad (2)$$

In these equations $N_{11111\,dCas9}$ represents the number of events with both the 11111 barcode a dCas9 bound. $N_{11111\,No\,dCas9}$ would represent the number of events with the 11111 barcode and no dCas9 bound.

The calculations of relative concentration have additional parameters because the total number of events without dCas9 bound for both barcoded nanostructures also plays a role. There is always some percentage of error with concentration experiments; thus, to account for that, the normalized bound percentage of accounts for the total number of events with a given barcode as measured in the nanopore. For the events in Fig. 2c,d, the following equations are used:

$$X = \frac{N_{11111\,dCas9}}{N_{11111\,dCas9} + N_{11001\,dCas9}} \times (N_{11111\,No\,dCas9} + N_{11111\,dCas9}) \qquad (3)$$

$$Y = \frac{N_{11001\,dCas9}}{N_{11111\,dCas9} + N_{11001\,dCas9}} \times (N_{11001\,No\,dCas9} + N_{11001\,dCas9}) \qquad (4)$$

$$\text{Normalized Bound}\%_X : \frac{X}{(X+Y)} \qquad (5)$$

$$\text{Normalized Bound}\%_Y : \frac{Y}{(X+Y)} \qquad (6)$$

The first part of this formula just looks at events with dCas9 bound $\frac{N_{11111\,dCas9}}{N_{11111\,dCas9} + N_{11001\,dCas9}}$, whereas the second part involves multiplying by the total number of events with the barcode being measured ($N_{11111\,No\,dCas9} + N_{11111\,dCas9}$). This normalizes the measurements based on the relative concentrations that are being measured in the nanopore.

For Fig. 3, a different nanostructure is used and values for normalized ratios are changed accordingly. Ratios are calculated the following ways:

$$X_{\text{Position1}} = \frac{N_{dCas9\,in\,Position1}}{N_{dCas9\,in\,Position1} + N_{NodCas9\,in\,Position1}} \qquad (7)$$

$$X_{\text{Position2}} = \frac{N_{dCas9\,in\,Position2}}{N_{dCas9\,in\,Position2} + N_{NodCas9\,in\,Position2}} \qquad (8)$$

$$X_{\text{Position3}} = \frac{N_{dCas9\,in\,Position3}}{N_{dCas9\,in\,Position3} + N_{NodCas9\,in\,Position3}} \qquad (9)$$

$$X_{\text{Control}} = \frac{\Sigma X_{\text{Control Position 1,2,3}}}{3} \qquad (10)$$

$$\text{Normalized Binding Ratio to Mismatched DNA}_{\text{Position }i} = \frac{X_{\text{Position }i}}{X_{\text{Control}}} \qquad (11)$$

This normalization ratio is slightly different and based on the binding efficiency of the target gRNA to its target DNA sequence. Each $X_{\text{Position }i}$ represents a different mismatch. $X_{\text{Control}}$ defined as $\frac{\Sigma X_{\text{Control Position 1,2,3}}}{3}$ is the average binding efficiency at each position of the target dCas9 to its target DNA sequence. This treats the measured ratio of the dCas9 RNP to its target as a binding ratio of 1.0 and the other measured ratios relative to that.

Binding efficiency (%) is calculated:

$$\text{Binding Efficiency to Target DNA sequence (\%)} = \frac{N_{dCas9}}{N_{dCas9} + N_{No\,dCas9}} \times 100 \qquad (12)$$

For the samples with multiple measurements, as described in the text, standard deviation of the population was calculated using the following:

$$\text{SD} = \sqrt{\frac{\Sigma(x - \bar{x})^2}{N}}$$

## Reporting summary

Further information on research design is available in the Nature Portfolio Reporting Summary linked to this article.

## Data availability

The main data supporting the results in this study are available within the paper and its Supplementary Information. The source raw data generated in this study are available at https://doi.org/10.17863/CAM.96534.

## Code availability

LabView 2016 was used for data acquisition. The CHOPCHOP (version 3) server is available at https://chopchop.cbu.uib.no. The Python code for local installation is available at https://bitbucket.org/valenlab/chopchop. Nanopore data analysis scripts are available at https://gitlab.com/keyserlab/nanopyre and https://gitlab.com/keyserlab/nanopro.

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

## Acknowledgements

S.E.S. acknowledges funding from Oxford Nanopore Technologies, Engineering and Physical Sciences Research Council (EPSRC) and Cambridge Trust. N.E.W. acknowledges funding from Oxford Nanopore Technologies, the Canada UK Foundation and the University of Cambridge Office of Postdoctoral Affairs. S.Y. acknowledges funding from the EPSRC (EP/S022953/1), and A.D. acknowledges funding from the EPSRC (EP/L015889/1). U.F.K. and K.C. acknowledge funding through the European Research Council (ERC-2019-POC PoreDetect 899538). We thank Z. Xuan and N. Ermann for assisting in the development of data analysis tools and C. Platnich for the helpful reading of the manuscript and useful suggestions.

## Author contributions

S.E.S., N.E.W., R.G. and U.F.K. conceived the idea. S.E.S., N.E.W. and K.C. designed the nanostructures. A.D. advised on dCas9 guide design. S.E.S. and S.Y. performed the experiments and analysed the nanopore data. S.E.S., N.E.W. and U.F.K. wrote the initial manuscript draft. All authors contributed to discussion and the final manuscript version.

## Competing interests

N.E.W. is a co-founder and consults for 52 North Health Ltd. R.G. is an employee of Oxford Nanopore Technologies. S.E.S. is partially funded by Oxford Nanopore Technologies for her PhD. The remaining authors declare no competing interests.

## Additional information

**Correspondence and requests for materials** should be addressed to Ulrich F. Keyser.

# Reporting Summary

## Statistics

For all statistical analyses, confirm that the following items are present in the figure legend, table legend, main text, or Methods section.

| n/a | Confirmed | |
|---|---|---|
| ☐ | ☒ | The exact sample size (*n*) for each experimental group/condition, given as a discrete number and unit of measurement |
| ☐ | ☒ | A statement on whether measurements were taken from distinct samples or whether the same sample was measured repeatedly |
| ☒ | ☐ | The statistical test(s) used AND whether they are one- or two-sided<br>*Only common tests should be described solely by name; describe more complex techniques in the Methods section.* |
| ☒ | ☐ | A description of all covariates tested |
| ☐ | ☒ | A description of any assumptions or corrections, such as tests of normality and adjustment for multiple comparisons |
| ☐ | ☒ | A full description of the statistical parameters including central tendency (e.g. means) or other basic estimates (e.g. regression coefficient) AND variation (e.g. standard deviation) or associated estimates of uncertainty (e.g. confidence intervals) |
| ☒ | ☐ | For null hypothesis testing, the test statistic (e.g. *F*, *t*, *r*) with confidence intervals, effect sizes, degrees of freedom and *P* value noted<br>*Give P values as exact values whenever suitable.* |
| ☒ | ☐ | For Bayesian analysis, information on the choice of priors and Markov chain Monte Carlo settings |
| ☒ | ☐ | For hierarchical and complex designs, identification of the appropriate level for tests and full reporting of outcomes |
| ☒ | ☐ | Estimates of effect sizes (e.g. Cohen's *d*, Pearson's *r*), indicating how they were calculated |

*Our web collection on statistics for biologists contains articles on many of the points above.*

## Software and code

Policy information about availability of computer code

| Data collection | LabView 2016 was used for data acquisition. The CHOPCHOP (version 3) server is available at https://chopchop.cbu.uib.no. The python code for local installation is available at https://bitbucket.org/valenlab/chopchop. |
|---|---|
| Data analysis | Nanopore data-analysis scripts are available at https://gitlab.com/keyserlab/nanopyre and https://gitlab.com/keyserlab/nanopro. |

For manuscripts utilizing custom algorithms or software that are central to the research but not yet described in published literature, software must be made available to editors and reviewers. We strongly encourage code deposition in a community repository (e.g. GitHub). See the Nature Portfolio guidelines for submitting code & software for further information.

## Data

Policy information about availability of data

All manuscripts must include a data availability statement. This statement should provide the following information, where applicable:
- Accession codes, unique identifiers, or web links for publicly available datasets
- A description of any restrictions on data availability
- For clinical datasets or third party data, please ensure that the statement adheres to our policy

The main data supporting the results in this study are available within the paper and its Supplementary Information. The source raw data generated in this study are available at https://doi.org/10.17863/CAM.96534.

## Human research participants

Policy information about studies involving human research participants and Sex and Gender in Research.

| | |
|---|---|
| Reporting on sex and gender | The study did not involve human research participants. |
| Population characteristics | — |
| Recruitment | — |
| Ethics oversight | — |

Note that full information on the approval of the study protocol must also be provided in the manuscript.

# Field-specific reporting

Please select the one below that is the best fit for your research. If you are not sure, read the appropriate sections before making your selection.

☒ Life sciences          ☐ Behavioural & social sciences          ☐ Ecological, evolutionary & environmental sciences

For a reference copy of the document with all sections, see nature.com/documents/nr-reporting-summary-flat.pdf

# Life sciences study design

All studies must disclose on these points even when the disclosure is negative.

| | |
|---|---|
| Sample size | Each nanopore device was run continuously until at least 1,000 DNA translocations occurred. The data was then filtered to remove folded and knotted events. There is no net sample size because the number of folded and knotted events and the noise levels vary, making the events unanalyzable. |
| Data exclusions | We only used unfolded DNA events with noise < 15 pA. |
| Replication | The specific replication of each experiment is described in the relevant figure captions and text. |
| Randomization | Randomization was not relevant to the study because we did not compare across experimental groups. |
| Blinding | Blinding was not relevant to the study because we did not compare across experimental groups. |

# Reporting for specific materials, systems and methods

We require information from authors about some types of materials, experimental systems and methods used in many studies. Here, indicate whether each material, system or method listed is relevant to your study. If you are not sure if a list item applies to your research, read the appropriate section before selecting a response.

### Materials & experimental systems

| n/a | Involved in the study |
|---|---|
| ☒ | Antibodies |
| ☒ | Eukaryotic cell lines |
| ☒ | Palaeontology and archaeology |
| ☒ | Animals and other organisms |
| ☒ | Clinical data |
| ☒ | Dual use research of concern |

### Methods

| n/a | Involved in the study |
|---|---|
| ☒ | ChIP-seq |
| ☒ | Flow cytometry |
| ☒ | MRI-based neuroimaging |

