## [Peer Review File · Nature Biomedical Engineering]

Sensing the DNA-mismatch tolerance of catalytically inactive Cas9 via barcoded DNA nanostructures in solid-state nanopores

Corresponding author: Ulrich Keyser

Editorial note

This document includes relevant written communications between the manuscript's corresponding author and the editor and reviewers of the manuscript during peer review. It includes decision letters relaying any editorial points and peer-review reports, and the authors' replies to these (under 'Rebuttal' headings). The editorial decisions are signed by the manuscript's handling editor, yet the editorial team and ultimately the journal's Chief Editor share responsibility for all decisions.

Any relevant documents attached to the decision letters are referred to as **Appendix #**, and can be found appended to this document. Any information deemed confidential has been redacted or removed. Earlier versions of the manuscript are not published, yet the originally submitted version may be available as a preprint. Because of editorial edits and changes during peer review, the published title of the paper and the title mentioned in below correspondence may differ.

Correspondence

Fri 20 Jan 2023

Decision on Article nBME-22-2556

Dear Prof Keyser,

Thank you again for submitting to *Nature Biomedical Engineering* your manuscript, "Nanopore sensing with DNA nanostructures reveals Guide-Intrinsic Mismatch Tolerance of CRISPR/dCas9". The manuscript has been seen by 3 experts, whose reports you will find at the end of this message.

You will see that the reviewers appreciate aspects of the work. However, they articulate concerns about the degree of support for some of the claims and provide useful suggestions for improvement. We hope that with significant further effort you can address the criticisms, increase the level of significance of the study, and convince the reviewers of its merits.

In particular, we would expect that a revised version of the manuscript provides:

- * Ideally, proof-of-concept evidence of the biomedical applicability of the assay. At minimum, discussion of the most suitable application(s) and of the current applicability limitations of the assay.
- * Evidence of the robustness and reproducibility of the experiments, as queried by Reviewer #2.
- * Additional controls, as per the relevant comments of Reviewers #2 and #3.
- * Thorough methodological reporting, as per the pertinent comments of all reviewers.

When you are ready to resubmit your manuscript, please upload the revised files, a point-by-point rebuttal to the comments from all reviewers, the reporting summary, and a cover letter that explains the main improvements included in the revision and responds to any points highlighted in this decision.Please follow the following recommendations:

- * Clearly highlight any amendments to the text and figures to help the reviewers and editors find and understand the changes (yet keep in mind that excessive marking can hinder readability).
- * If you and your co-authors disagree with a criticism, provide the arguments to the reviewer (optionally, indicate the relevant points in the cover letter).
- * If a criticism or suggestion is not addressed, please indicate so in the rebuttal to the reviewer comments and explain the reason(s).
- * Consider including responses to any criticisms raised by more than one reviewer at the beginning of the rebuttal, in a section addressed to all reviewers.
- * The rebuttal should include the reviewer comments in point-by-point format (please note that we provide all reviewers will the reports as they appear at the end of this message).
- * Provide the rebuttal to the reviewer comments and the cover letter as separate files.

We hope that you will be able to resubmit the manuscript within 25 weeks from the receipt of this message. If this is the case, you will be protected against potential scooping. Otherwise, we will be happy to consider a revised manuscript as long as the significance of the work is not compromised by work published elsewhere or accepted for publication at *Nature Biomedical Engineering*.

We hope that you will find the referee reports helpful when revising the work. Please do not hesitate to contact me should you have any questions.

Best wishes,

Valeria

Dr Valeria Caprettini
Associate Editor, Nature Biomedical Engineering

Reviewer #1 (Report for the authors (Required)):

Summary: In this work, the authors develop a new nanopore sensing technology that uses DNA nanostructures for the detection of CRISPR/dCas9 binding. Briefly, this system uses synthetic DNA sequences comprised of a barcode region (containing DNA 'dumbbell' like structures), and an upstream dCas9 binding site encoded by a target DNA nanostructure. This strand is subject to nanopore sequencing in the absence or presence of dCas9 and a guide RNA. As dsDNA is read through the nanopore, an ionic current drop in signal occurs; when an analyte such as a protein binds, a secondary spike occur. Through decoding methods described in the paper, changes in mA associated with reading the strand can be translated into an ID of the particular sequence (a binary 5-bit identifier), and the presence or absence of bound dCas9. The authors validate the specificity of this technique as well as its ability to quantify DNA-protein interactions. In addition, they apply this technique to examining how single nucleotide changes in the target DNA affect binding of the complex. Finally, they also examine the effect of guide intrinsic mismatch on binding, and discuss the possibility of using multiple arrayed binding sites (DNA nanostructures for dCas9 docking). Overall, this is a novel new system which builds on previous work by this group. It could potentially be very useful for certain applications, but these do need to be better explained and developed in the current paper. I would suggest that it would be suitable for publication in *Nature Biomedical Engineering* if the authors are able to address the points listed below:

Minor points:

- The manuscript is generally well written but there are still some minor mistakes
- should include page and line numbers
- Spacing in a few places: e.g. 2 To Capture the full potential... (too many spaces around the 2).
- legend on Fig. 1 (theschematic  needs a space in between)
- as seen in Fig. 2D (period)
- Labelling for Fig. 3B is very confusing. Is 3B missing?
- Inconsistent spacing – e.g. references 35 and 36; one has zero spaces, the other has one.

-The authors cite two papers on previous work using nanopore sequencing and dCas9 (2, 13). While reading the abstracts of these suggests the current work is quite distinct and advances the field, the authors may want to include a sentence in the paper about why this work is different than the other two.

-Statistics are absent from the figure legends. While this reviewer understands that each graph represents multiple events, were the entire experiments performed multiple times, or was $n=1$? This should be stated.

-Error bars when present are not defined – e.g. 4B – standard deviation, standard error? $N=?$

Major points:

-Can the barcode be increased in length and to what extent? 2^5 would only allow for 32 different sequences to be resolved

-What is typical in terms of event numbers for these experiments? perhaps the authors should cite some precedence, etc. or explain why $n=60$ is sufficient; also, can stats be someone derived and included on graphs for this (e.g. Fig. 2)? How was this number of events determined beforehand?

-Data presented in Figure 3 is very confusing. It is unclear what the library size used to generate the heatmap was. Was every base pair mutation tested in every position? If so, this would seem to be much larger than the available number of barcodes. The exact library of target sequences that was tested should be clearly indicated (how many sequences) and described in the text. It is unclear if it is just the 3 in A, or if all combinations were tested, or anything in-between. This was done with one crRNA I presume? This needs to be much better labelled in the figure and explained in the text. It is unclear what the heatmap represents or what data was used to derive it. The minimum library to achieve the heatmap in C would presumably be 4^{17} , but they only collect 60 events. Maybe this isn't the best way to present these results?

-I don't think heat maps are appropriate for Fig. 4C or 4F. I see what the authors are trying to present, but having a heatmap here makes the data very confusing. Perhaps just make these into a bar graph showing the relative DNA/RNA change at each position (dividing the values). Or stating in the text that the DNA bp change is X-fold greater than when an RNA change was made at that position.

-In relation to the discussion of wobble base pairs, researchers have shown that incorporation of inosine into dCas9 crRNAs reduces binding to the cognate DNA sequence (see/cite Kryslar et al., Nature Comm., 2022) but allows for pairing with sequences bearing single substitutions at overlapping positions. This supports the author's assertion that wobble base pairing could be the cause for the observed specificity results. Moreover, the structural basis for these wobble effects has recently been established in the context of Cas9 (see/cite Pacesa et al., Cell, 2022).

-I would like to see, for at least one of these data presented, a reproduced experiment just to get an idea of how consistent the results are with this technique

-There are definitely applications for this system but they need to be discussed in more detail. Due to the apparent limitation of the size of libraries being barcoded, I'm not sure this is the best tool for evaluating the specificity of a particular probe (there are established in vitro and cell-based high-throughput techniques for this such as SELEX and CHIP-Seq). This technology could however, be potentially be used as an alternative diagnostic system (for example if one could derive the guide RNA out of a test sample and then run it in the presence of Cas9 through the Nanopore sequencer). But this should or similar possibilities should be detailed more in depth. There would also be applications for this in the area of drug discovery  looking for dCas9 inhibitors, and other assays where traditionally an EMSA would be used. The idea of quantifying mutations in a sample seems like an appropriate use of the technology, if the advantages over sequencing are clearly explained.

Reviewer #2 (Report for the authors (Required)):

The manuscripts titled “Nanopore Sensing with DNA Nanostructures Reveals Guide-Intrinsic Mismatch Tolerance of CRISPR/dCas9” described a solid state nanopores with diameters of ~10 nm to identify binding events between DNA and CRISPR associated (Cas) probes. The reported system contains designed DNA nanostructures, which allows for the incorporation of user-defined binding sequences for a systematic study of how mismatch position impacts the binding efficiency. The results presented here revealed the relationship between sequence and binding at the single nucleotide level by using nanopore based measurements and DNA nanotechnology for biosensing.

Comment#1 In Figure1 Four different combination of experiment performed with (a) dCas9 with 11111 probe; (b) dCas9 with 11001; (c) DNA probe with 11111; (d) DNA probe with 11001. To further clarify that signal in combination (a) and (b) due to dCas9 binding with hanging DNA probe not just random binding or just presence or absence of dCas9 in the solution, a control experiment needs to be performed with non-target DNA probe (without PAM and target site).

Comment#2: as DNA probe contain multiple component assembly. how much is the yield of the proposed probe. It is also very strange to see in the result section that dumbbell shaped probe in one hand provide peaks while hanging larger dsDNA probe signal almost negligible. Could you please explain it.

Comment#3 The reported work has interesting probe design, use of 10 nm size nanopipette for evaluating binding of dCas9. However, it is very hard to find any new information in terms of CRISPR-Cas9 and target site binding, role of gRNA and mismatch in target site.

Comment #4 The normalization processes used for half of the data are not thoroughly explained. The rationale for the normalization is clear, but the formulas/equations they provide are not at all intuitive. The language is not consistent between the equations in the methods section (Analysis of Nanopore Data) and the axes on the figures in the main text, which makes it even harder to follow. Additionally, the authors quantify the number of occurrences for each event type but fail to provide those numbers in detail anywhere. A table of values for N in the SI or something for at least the data presented in Figure 2 is required.

Reviewer #3 (Report for the authors (Required)):

Please see attached pdf document.

Appendix 1

Wed 05 Apr 2023

Decision on Article nBME-22-2556A

Dear Prof Keyser,

Thank you for your revised manuscript, "Nanopore sensing with DNA nanostructures reveals Guide-Intrinsic Mismatch Tolerance of CRISPR/dCas9". Having consulted with the original reviewers (whose comments you will find at the end of this message), I am pleased to write that we shall be happy to publish the manuscript in *Nature Biomedical Engineering*.

We will be performing detailed checks on your manuscript, and in due course will send you a checklist detailing our editorial and formatting requirements. You will need to follow these instructions before you upload the final manuscript files.

Best wishes,

Valeria

Dr Valeria Caprettini
Associate Editor, Nature Biomedical Engineering

Reviewer #1 (Report for the authors (Required)):

The authors have addressed all of my comments.

Reviewer #2 (Report for the authors (Required)):

Thank you. The manuscript have improved and all of my comments have been addressed.

Reviewer #3 (Report for the authors (Required)):

I have reviewed the manuscript after the major revision and I am happy to report that the authors have addressed all of the concerns raised in my previous review. The manuscript has been significantly improved and I would recommend its acceptance for publication.

Appendix 1

This work combines the barcoded nanostructures, nanopore reading and CRISPR/dCas system to demonstrate a versatile system with capability of (1) determining relative concentration of multiple targets in a mixture by using a single reaction and a single readout; (2) high-throughput test of multiple gRNAs design in a single measurement, and (3) examining the base mismatch effect on binding efficiency. This work is built upon the author's previous protocol on barcoded carrier DNAs. I am quite excited to see its expansion to evaluating the CRISPR/dCas system.

To start, the general findings of this work are as follows.

1. A barcode DNA dumbbell region could identify different target DNA by showing featured spikes in event traces.
2. The deeper current spike pattern in event traces is the fingerprint of DNA-dCas9 binding.
3. The binding efficiency between target DNA and dCas9 could be evaluated by the ratio of events with a binding spike in all events detected.
4. The system could differentiate two mixed DNA targets, and quantify their relative abundance.
5. Single nucleotide mismatch of target DNA will affect the DNA-dCas9 binding specificity with mismatch position and base change. The data showed a lower specificity when the mismatch was located at PAM distal region.
6. The study of single nucleotide mutation of gRNA also suggests that binding specificity is affected by the mutation position. The data showed a lower specificity when the mutation was located at PAM distal region.

These conclusions indeed showed that this system with specially designed DNA nanostructures could be used for assessing dCas9 binding efficiency and multiplexed detection. The results showed good quantification and specificity of the system. Compared with computational tools for predicting cleavage, the nanopore readout can directly and quantitatively assess the binding between target DNA and dCas9. This direct assessment could guide the design of dCas9 RNPs.

With this excitement, I have many questions that need to be clarified.

1. Is there negative control (no DNA/protein added in solution) trace? What is the typical threshold value of the current blockade to find an event?
2. Any comments on extending this technology to other Cas proteins? Considering most Cas proteins have cleavage activity (rather than the dCas studied here), how does the mismatched effect conclusion extend to other systems?
3. The 2M/4M LiCl is used for nanopore measurement. Fig. S5 showed lower binding efficiency at a higher salt concentration. Will the binding efficiency reduce when dCas9 proteins stay longer in the high salt solution? Could the binding efficiency measured under high salt conditions predict the real binding efficiency under normal buffer conditions for normal Cas proteins? For example, NEB 3.1 buffer for Cas9 contains only 100 mM NaCl and it's well known that Cas9 would lose its activity when working at high salt concentrations like 2M/4M LiCl.
4. In Fig. 3A, the highlighted position seems to be 16 nt away from the PAM instead of 14.
5. In Fig. 3C, does the GACCACUACCGA above the heatmap indicate the gRNA sequence? How could the target base pair have the highest binding ratio for combinations of dG-rG, dC-rC, dT-rU, dA-rA? Shouldn't dG-rC, dC-rG, dT-rA, dA-rU be the highest (almost value of 1)?
6. On page 8, the 2nd paragraph says: "For all experiments, N>45 clearly labelled distinguishable events." Does this mean 45 events were enough for analysis? Would this suffer from significant Poisson noise?
7. In the caption of Fig. 4, labels like (a), (b), and (c) should be capitalized to keep consistency.

8. In Fig. 4A, for probe 2, gRNA is labeled in DNA format, which should be UCAUGUCUUACCCCAAUAA. Same issue for Fig. 4D.
9. In Fig. 4F, is the binding ratio data from Fig. 4E? It seems to be not consistent with each other. For mutation in positions 1 to 4, the ratio is 0.2, 0.2, 0.3, 0.8 in Fig. 4E, and 0.2, 0.4, 0.1, 0.8 in Fig. 4F.
10. On page 10, the middle of 1st paragraph says: "Probe 2 showed to have very little ... sensitivity to single base pair mutations ... this lack of specificity highlights that this probe would not be useful as a tool for detecting single base pair changes in clinical samples." Then why choose Probe 2 for the mutation study in Fig. 4D, E, F?
11. On page 12, the "Nanopore Fabrication and Measurement" part says: "The measurement is then taken until around 1000 folded and unfolded events are gathered". How to define the unfolded events, and what do they look like?
12. On page 13, the end of 1st paragraph says: "Following the sorting, the events are analysed by eye and events which have folds or knots interfering with the barcode are discarded." Is there any objective way to inspect events instead of manually checking?
13. In equation (10), what does n mean?

Rebuttal 1

Response to reviewers on “Nanopore Sensing with DNA Nanostructures Reveals Guide-Intrinsic Mismatch Tolerance of CRISPR/dCas9”

We thank the reviewers for their careful reading and constructive comments. We addressed their comments and made changes to the revised manuscript, as detailed below.

[Reviewers comments are italicized; our response is normal black text; our changes to the revised ms. are blue text] and listed below the corresponding comments.

Reviewer #1:

Minor points:

- The manuscript is generally well written but there are still some minor mistakes*
- should include page and line numbers*
- Spacing in a few places: e.g. 2 To Capture the full potential... (too many spaces around the 2).*
- legend on Fig. 1 (theschematic  needs a space in between)*
- as seen in Fig. 2D (period)*
- Labelling for Fig. 3B is very confusing. Is 3B missing?*
- Inconsistent spacing – e.g. references 35 and 36; one has zero spaces, the other has one.*

All of the above has been changed

-The authors cite two papers on previous work using nanopore sequencing and dCas9 (2, 13). While reading the abstracts of these suggests the current work is quite distinct and advances the field, the authors may want to include a sentence in the paper about why this work is different than the other two.

A sentence was added to the text expanding on this (Line 65, Page 2):

‘Our system builds upon these initial proof of concept works by introducing designed DNA nanostructures for user-defined DNA target testing, thus expanding the application space of nanopore sensing, DNA nanotechnology and screening of protein-DNA interactions.’

-Statistics are absent from the figure legends. While this reviewer understands that each graph represents multiple events, were the entire experiments performed multiple times, or was $N=1$? This should be stated.

We have now included the number (N) of events and tests in each caption of the figures in the updated manuscript.

-Error bars when present are not defined – e.g. 4B – standard deviation, standard error? $N=?$

This was standard deviation and is now included in the figure caption.

Line 297, Page 10:

“(B) Comparing binding efficiency of probes to each other and to the predicated efficiency from CHOPCHOP (grey)¹. $N=288$ events for probe 1 and $N=558$ events for probe 2. Error bars are the standard deviation between at least four different measurements in different pores for each probe.”

Major points:

-Can the barcode be increased in length and to what extent? 2^5 would only allow for 32 different sequences to be resolved

The number of barcodes can indeed be easily increased by adding more bits. We have shown that 56 bits are possible in a previous work², meaning that we could encode 2^{56} types of barcodes in our carrier. The limitation here is the generation of these barcodes. The expansion of barcodes is now being discussed in the manuscript.

Line 136, Page 4:

“The barcode design is based on previous work which was optimized to create clearly distinguishable spikes in nanopores with diameters around 15 nm⁷. While this barcode design leaves us with only 2^5 (32) sequences that can be tested, we have already explored the maximum limits of barcodes. We can increase the density of the nanostructures and the signal-to-noise ratio using smaller nanopores with 56 bits on a single the DNA carrier, allowing for a library of 2^{56} ($> 10^{16}$) molecules.²⁵ The number of multiplexed protein-DNA interactions is only limited by the number of nanopores and our ability to assemble these DNA nanostructures.”

-What is typical in terms of event numbers for these experiments ? perhaps the authors should cite some precedence, etc. or explain why $N=60$ is sufficient; also, can stats be someone derived and included on graphs for this (e.g. Fig. 2)? How was this number of events determined beforehand?

In a previous work, *Zhu et al.* perform calculations to determine the maximum number of events needed in the supplementary of their work ‘Image Encoding Using Multi-Level DNA Barcodes with Nanopore Readout’. The number of events for every experiment has been added to the figure captions.

Additionally, the following has been added in the Analysis of Nanopore data in the methods section (Line 440, Page 14):

“Our lab has shown that as few as 4 events are sufficient for positive detection in the majority of cases, while 9 correct events increase the probability of positive detection to more than 90%³.

- Data presented in Figure 3 is very confusing. It is unclear what the library size used to generate the heatmap was. Was every base pair mutation tested in every position? If so, this would seem to be much larger than the available number of barcodes. The exact library of target sequences that was tested should be clearly indicated (how many sequences) and described in the text. It is unclear if it is just the 3 in A, or if all combinations were tested, or anything in-between. This was done with one crRNA I presume? This needs to be much better

labelled in the figure and explained in the text. It is unclear what the heatmap represents or what data was used to derive it. The minimum library to achieve the heatmap in C would presumably be 4^{17} , but they only collect 60 events. Maybe this isn't the best way to present these results?

The authors have found this comment to be extremely useful and Figure 3 has been reformatted for clarification. The referee is correct that the crRNA sequence was the same. We chose to use one crRNA here as in sensing applications one would use a dedicated crRNA to screen the target DNA for the desired sequence.

Line 232, Page 8:

Fig. 3. Measuring sensitivity to single base pair changes. (A) The dCas9 RNP used for assessing specificity to mutations. (B) Control nanostructure with target sequence without any mutations in overhang region. (C) Nanostructure with overhangs with mismatches in the target DNA overhang region. (D) Schematic of Nanostructure showing dCas9 binding to overhang and corresponding nanopore event trace. The furthest on the left represents binding of the RNP to a nanostructure with three overhangs which are control. The middle event represents the three mismatched overhangs with no observed binding of the RNP. The furthest of the right represents the dCas9 RNP binding to two of the mismatched target sequences, highlighting low specificity. (E) Bar plots for each possible different DNA base mutation and position with Normalized binding ratio being calculated using equations (7-11). $N > 45$ events for each experiment. Error bars are result of standard deviation from normalization to control binding efficiency.

The heatmap was removed and replaced with bar charts directly indicating the normalised binding ratio. Every base pair was tested in every position with three mismatches tested per experiment. This is a total of 16 experiments for testing the mismatched DNA, each with $N > 45$, each of which the results are included in a bar plot in Figure 3E. Additionally, a

control nanostructure with target DNA at every overhang site was tested and incorporated into the figure and then the binding efficiency of this was shown in in Figure S3.

Figure S3. Binding efficiency of the dCas9 probe in Figure 3 to the different positions on the DNA nanostructures in Figure 3.

The calculation for normalization was also expanded upon. There is a total of 48 different sequences tested which has now been mentioned in the text (Line 249, Page 8): The sequences for these overhangs can be found in Table S4. A single crRNA was tested.

-I don't think heat maps are appropriate for Fig. 4C or 4F. I see what the authors are trying to present, but having a heatmap here makes the data very confusing. Perhaps just make these into a bar graph showing the relative DNA/RNA change at each position (dividing the values). Or stating in the text that the DNA bp change is X-fold greater than when an RNA change was made at that position.

The authors agree with this comment, find it very helpful and have now reformatted it to be clearer by using bar graphs.

Line 295, Page 10:

Fig. 4. Probe-to-Probe variation on specificity of RNP-complex. (A) Probe 1 and Probe 2 were designed and compared using nanostructures described in Fig. 1. (B) Comparing binding efficiency of probes to each other and to the predicted efficiency from CHOPCHOP (grey)¹. $N=288$ events for probe 1 and $N=558$ events for probe 2. Error bars are the standard deviation between at least four different measurements in different pores for each probe. (C) Bar graphs of binding ratio normalized to efficiency of probes to target. Sensitivity to single base pair changes varies greatly between the two probes, highlighting influence of guide-intrinsic mismatch tolerance (GMT). Each experiment has at least $N>45$ events. (D) Variations of Probe 2 with mutations in gRNA (E) Comparison between predicted efficiency (grey) and measured binding efficiency when mutations are introduced into gRNA. Each experiment has at least greater than $N>55$ events. (f) Bar graphs with normalized binding ratios depicting probe-dependence of GMT and large variation of binding among probes with introduced mutations. Each experiment has at least greater than $N>55$ events.

-In relation to the discussion of wobble base pairs, researchers have shown that incorporation of inosine into dCas9 crRNAs reduces binding to the cognate DNA sequence (see/cite Kryslar et al., Nature Comm., 2022) but allows for pairing with sequences bearing single substitutions at overlapping positions. This supports the author's assertion that wobble base pairing could be the cause for the observed specificity results. Moreover, the structural basis for these wobble effects has recently been established in the context of Cas9 (see/cite Pacesa et al., Cell, 2022).

This has been included now in the discussion about wobble base pairing (Page 9, Line 278): The theory that the ‘wobble’ base pairing influences specificity agrees with recent works where the incorporation of inosine into dCas9 gRNAs reduces binding to cognate DNA sequences, but allows for pairing with sequences bearing single substitutions at overlapping position⁴. Additionally, the structural basis of wobble effects in the context of Cas9 has been established in the literature, supporting these claims⁵.

-I would like to see, for at least one of these data presented, a reproduced experiment just to get an idea of how consistent the results are with this technique

We have done these experiments. However, in response to the confusion of the referee we have now updates Figure 2D to visualize the three independent repeats. This figure was now further labelled to clarify that these bars represent three individual repeats.

Line 178, Page 6:

Fig. 2. Quantifying DNA binding to nanostructures. A) dCas9 probes and DNA nanostructures overhang schematics depicting single base difference between sequences in crRNA probe (with the PAM in red) and DNA overhang. The single base pair change can be found in Kat315 gene in Isoniazid-resistant *Mycobacterium Tuberculosis*. Two different DNA nanostructures with different overhangs (green and purple) are depicted. B) Binding specificity of the probes when both DNA nanostructures are added and only one respective probe is added. $N > 65$ events for each experiment. A control nanostructure with target DNA containing no PAM was also tested as seen in grey, where $N > 165$. Calculations were made using equation (1) and (2). C) Quantification of labelled events dependent on varying relative concentration ratio of DNA

nanostructures (green or purple) in solution with equimolar concentrations of the two dCas9 probes (standard deviation represents one sample measured in at least two different pores). Calculations were made using equation (3-6). $N > 85$ events for each condition. D) Reproduced experiments with 11001 and 11111 nanostructures barcode at a ratio of 2:1 in solution with equimolar concentrations of both dCas9 probes added. The different bars (1,2,3) are three independent sample preparation repeats in three different nanopores. Calculations were made using equation (3-6). $N > 100$ events for each measurement. In all experiments, dCas9 probes were added in excess to target DNA molecules.

-There are definitely applications for this system but they need to be discussed in more detail. Due to the apparent limitation of the size of libraries being barcoded, I'm not sure this is the best tool for evaluating the specificity of a particular probe (there are established in vitro and cell-based high-throughput techniques for this such as SELEX and ChIP-Seq). This technology could however, be potentially be used as an alternative diagnostic system (for example if one could derive the guide RNA out of a test sample and then run it in the presence of Cas9 through the Nanopore sequencer). But this should or similar possibilities should be detailed more in depth. There would also be applications for this in the area of drug discovery  looking for dCas9 inhibitors, and other assays where traditionally an EMSA would be used. The idea of quantifying mutations in a sample seems like an appropriate use of the technology, if the advantages over sequencing are clearly explained.

The author find this suggestion extremely useful, and have expanded on the applications beyond the high throughput nature of the device in the discussion (Line 356, Page 12) :

Our methodology has the potential to be translated to applications beyond those shown in this work in both diagnostics and gene-editing. With an expanded set of barcodes, it can be used as a highly specific, high-throughput approach to assaying the dCas9 RNPs to test hundreds of gRNAs in the same measurement. While there are in-vitro and cell based high-throughput techniques, such as ChIP-Seq⁶, it may be excessive depending on the final application. One advantage of the nanopore system is a few sequences of interest can be tested with minimal sample amounts (fMol per experiment), which can save the user time and resources. A reduction of enzyme amounts as well as target samples is especially relevant in the diagnostic

use cases^{7,8} and investigations of dCas9 inhibition⁹, where electrophoresis mobility shift assay (EMSA) are the de-facto standard. In terms of developing diagnostics, using dCas9 as a label on native DNA has already been demonstrated¹⁰ and by using the nanopore DNA-nanostructure system one can efficiently design and test dCas9 probes.

Reviewer #2 (Report for the authors (Required)):

The manuscripts titled “Nanopore Sensing with DNA Nanostructures Reveals Guide-Intrinsic Mismatch Tolerance of CRISPR/dCas9” described a solid state nanopores with diameters of ~10 nm to identify binding events between DNA and CRISPR associated (Cas) probes. The reported system contains designed DNA nanostructures, which allows for the incorporation of user-defined binding sequences for a systematic study of how mismatch position impacts the binding efficiency. The results presented here revealed the relationship between sequence and binding at the single nucleotide level by using nanopore based measurements and DNA nanotechnology for biosensing.

Comment#1 In Figure1 Four different combination of experiment performed with (a) dCas9 with 11111 probe; (b) dCas9 with 11001; (c) DNA probe with 11111; (d) DNA probe with 11001. To further clarify that signal in combination (a) and (b) due to dCas9 binding with hanging DNA probe not just random binding or just presence or absence of dCas9 in the solution, a control experiment needs to be performed with non-target DNA probe (without PAM and target site).

An additional experiment was performed with non-target DNA which contained no PAM site. This was added both into the figure (Page 6, Line 178):

Fig. 2. Quantifying DNA binding to nanostructures. A) dCas9 probes and DNA nanostructures overhang schematics depicting single base difference between sequences in crRNA probe (with the PAM in red) and DNA overhang. The single base pair change can be found in Kat315 gene in Isoniazid-resistant *Mycobacterium Tuberculosis*. Two different DNA nanostructures with different overhangs (green and purple) are depicted. B) Binding specificity of the probes when both DNA nanostructures are added and only one respective probe is added. $N > 65$ events for each experiment. A control nanostructure with target DNA containing no PAM was also tested as seen in grey, where $N > 165$. Calculations were made using equation (1) and (2). C) Quantification of labelled events dependent on varying relative concentration ratio of DNA nanostructures (green or purple) in solution with equimolar concentrations of the two dCas9 probes (standard deviation represents one sample measured in at least two different pores). Calculations were made using equation (3-6). $N > 85$ events for each condition. D) Reproduced experiments with 11001 and 11111 nanostructures barcode at a ratio of 2:1 in solution with equimolar concentrations of both dCas9 probes added. The different bars (1,2,3) are three independent sample preparation repeats in three different nanopores. Calculations were made using equation (3-6). $N > 100$ events for each measurement. In all experiments, dCas9 probes were added in excess to target DNA molecules.

This was also described in the text (Line 169, Page 5): The nanostructures with a mutation that mismatched to the probe being tested were found to have binding of 5.3% and 6% respectively. Similarly, a control done with no PAM sequence and the probe above was found to have a binding of 4.2% ($N=169$). These false positives are most likely due to knots in the DNA during translocation which may produce similar signals to a bound dCas9 protein in the event trace.

Comment#2: as DNA probe contain multiple component assembly. how much is the yield of the proposed probe. It is also very strange to see in the result section that dumbbell shaped probe in one hand provide peaks while hanging larger dsDNA probe signal almost negligible. Could you please explain it.

Information was added into the methods DNA nanostructure barcodes section describing the yield (Page 12, Line 382): The sample is then filtered using a 100 kDa Amicon filter and measured in a nanodrop spectrophotometer for concentration information. Based on the nanodrop measurement, typical yield is 75-95%.

The final concentration of M13 backbone is 20 nM before putting it in the thermocycler and typically the concentration of the DNA nanostructure after Amicon filtering is 15-19 nM.

Additional information describing the design concerning the dumbbells and dsDNA overhang were added to the text to clarify why the peak from the dumbbells is observed but the dsDNA strand is not:

For the nanostructures in Fig. 1, within the 190 oligos are five groups of equally spaced simple dumbbell hairpin motifs to create the spikes which act as a barcode on the DNA nanostructure⁷. Each group consists of 11 DNA dumbbells to create a single spike. The exact sequences with their numbers are shown in Table S1 in the Supplementary Information following a previous work⁷. The overhang was created by replacing oligos No. 142 and 143 with a 90bp oligo made up of 30 bp segments to match the M13 backbone and 50 bp of the specific sequences containing the target sequences we aimed to test. The 50bp dsDNA overhang is not large enough to generate a current blockade which can be observed.

Comment#3 The reported work has interesting probe design, use of 10 nm size nanopipette for evaluating binding of dCas9. However, it is very hard to find any new information in terms of CRISPR-Cas9 and target site binding, role of gRNA and mismatch in target site.

This work builds upon previous works and demonstrates a new technique for verifying probes. The potential applications for which this can be used have been expanded upon in the discussion.(Line 356, Page 12) :

Our methodology has the potential to be translated to applications beyond those shown in this work in both diagnostics and gene-editing. With an expanded set of barcodes, it can be used as a highly specific, high-throughput approach to assaying the dCas9 RNPs to test hundreds of gRNAs in the same measurement. While there are in-vitro and cell based high-throughput techniques, such as ChIP-Seq⁶, it may be excessive depending on the final application. One advantage of the nanopore system is a few sequences of interest can be tested with minimal sample amounts (fMol per experiment), which can save the user time and resources. A reduction of enzyme amounts as well as target samples is especially relevant in the diagnostic use cases^{7,8} and investigations of dCas9 inhibition⁹, where electrophoresis mobility shift assay (EMSA) are the de-facto standard. In terms of developing diagnostics, using dCas9 as a label on native DNA has already been demonstrated¹⁰ and by using the nanopore DNA-nanostructure system one can efficiently design and test dCas9 probes.

Comment #4 The normalization processes used for half of the data are not thoroughly explained. The rationale for the normalization is clear, but the formulas/equations they provide are not at all intuitive. The language is not consistent between the equations in the methods section (Analysis of Nanopore Data) and the axes on the figures in the main text, which makes it even harder to follow. Additionally, the authors quantify the number of occurrences for each event type but fail to provide those numbers in detail anywhere. A table of values for N in the SI or something for at least the data presented in Figure 2 is required.

In order to make the normalization process more clear, schematic with the control nanostructure were introduced into Figure 3A and 3D. Additionally the process was expanded on in the text (Line 173, Page 5):

The equations used to calculate normalized binding ratio can be found in the methods section. These calculations rely on normalizing to the value for binding to the control (Xcontrol (10))

which was 33.7% with a 2.4% standard deviation as seen in Figure S3. Thus, if a probe is found to have a binding percentage of 33.7%, it would correspond to a normalized binding ratio of 1.0. The normalization is an essential step because this lower binding efficiency for these experiments is due to an increase in salt concentration in the buffer. With more DNA dumbbells, a higher salt concentration must be used to slow translocation for the nanopore, as seen in Fig. S4, however, this does reduce efficiency of binding as seen in Fig. S5.

Additionally following the equations the following text was added (Line 445, Page 14):

In these equations $N_{11111\ dCas9}$ represents the number of events with both the 11111 barcode and dCas9 bound. $N_{11111\ No\ dCas9}$ would represent the number of events with the 11111 barcode and no dCas9 bound.

The first part of this formula just looks at events with dCas9 bound $\frac{N_{11111\ dCas9}}{N_{11111\ dCas9} + N_{11001\ dCas9}}$, whereas the second part involves multiplying by the total number of events with the barcode being measured ($N_{11111\ No\ dCas9} + N_{11111\ dCas9}$). This normalizes the measurements based on the relative concentrations that are being measured in the nanopore.

Additionally, in the formulas for X and Y, N_{11111} was changed to $N_{11111\ No\ dCas9}$ and $X_{Control} = \frac{\sum X_i}{n}$ was changed to $X_{Control} = \frac{\sum X_{Control\ Position\ 1,2,3}}{3}$. The normalized ratio used for this set of experiments was also expanded on in the text (Page 15, Line 469):

This normalization ratio is slightly different and based on the binding efficiency of the target DNA to its target DNA sequence. Each $X_{Position\ i}$ represents a different mismatch. $X_{Control}$ defined as $\frac{\sum X_{Control\ Position\ 1,2,3}}{3}$ is the average binding efficiency at each position of the target dCas9 to its target DNA sequence. This treats the measured ratio of the dCas9 RNP to its target as a binding ratio of 1.0 and the other measured ratios relative to that.

And the Binding Efficiency % calculation was added:

Binding Efficiency (%) is calculated:

$$Binding\ Efficiency\ to\ Target\ DNA\ sequence(\%) = \frac{N_{dCas9}}{N_{dCas9} + N_{No\ dCas9}} * 100 \quad (12)$$

The axes in the main text were also altered to reflect the text used to describe the calculations. In reference to the Ns not being provided, after it was selected to be sent out for review and the nature guidelines were sent, updates were made that must not have been to sent to reviewers. In this most recent version, one can see one of the major changes which was including the N in each caption.

Reviewer 3:

This work combines the barcoded nanostructures, nanopore reading and CRISPR/dCas system to demonstrate a versatile system with capability of (1) determining relative concentration of multiple targets in a mixture by using a single reaction and a single readout; (2) high-throughput test of multiple gRNAs design in a single measurement, and (3) examining the base mismatch effect on binding efficiency. This work is built upon the author's previous protocol on barcoded carrier DNAs. I am quite excited to see its expansion to evaluating the CRISPR/dCas system.

To start, the general findings of this work are as follows.

- 1. A barcode DNA dumbbell region could identify different target DNA by showing featured spikes in event traces.*
- 2. The deeper current spike pattern in event traces is the fingerprint of DNA-dCas9 binding.*
- 3. The binding efficiency between target DNA and dCas9 could be evaluated by the ratio of events with a binding spike in all events detected.*
- 4. The system could differentiate two mixed DNA targets, and quantify their relative abundance.*
- 5. Single nucleotide mismatch of target DNA will affect the DNA-dCas9 binding specificity with mismatch position and base change. The data showed a lower specificity when the mismatch was located at PAM distal region.*
- 6. The study of single nucleotide mutation of gRNA also suggests that binding specificity is affected by the mutation position. The data showed a lower specificity when the mutation was located at PAM distal region.*

These conclusions indeed showed that this system with specially designed DNA nanostructures could be used for assessing dCas9 binding efficiency and multiplexed detection. The results showed good quantification and specificity of the system. Compared with computational tools for predicting cleavage, the nanopore readout can directly and quantitatively assess the binding between target DNA and dCas9.

This direct assessment could guide the design of dCas9 RNPs.

With this excitement, I have many questions that need to be clarified.

- 1. Is there negative control (no DNA/protein added in solution) trace? What is the typical threshold value of the current blockade to find an event?*

Figure S1 has now been updated to include the negative control trace where there is no DNA/protein being measured, additionally, on the raw trace, now an event is highlighted and included.

2. *Any comments on extending this technology to other Cas proteins? Considering most Cas proteins have cleavage activity (rather than the dCas studied here), how does the mismatched effect conclusion extend to other systems?*

The authors added some comments addressing the outlook of translating the system to one that can measure other Cas proteins into the discussion (Page 11, Line 344):

While currently, this system is limited to detection of dCas9 binding, there is potential for translation as a tool to measure cleavage as well. It has been shown that SpCas9 remains bound very stably to DNA after cleavage³⁹. Similarly, Cas12a cleaves the PAM distal region of the target, but remains bound in the PAM proximal.⁴⁰ There is potential to innovate new DNA nanostructure designs similar to the one discussed in this work and test how the cleavage of a variety of different enzymes can be affected by mismatches in the target DNA.

3. *The 2M/4M LiCl is used for nanopore measurement. Fig. S5 showed lower binding efficiency at a higher salt concentration. Will the binding efficiency reduce when dCas9 proteins stay longer in the high salt solution? Could the binding efficiency measured under high salt conditions predict the real binding efficiency under normal buffer conditions for normal Cas proteins? For example, NEB 3.1 buffer for Cas9 contains only 100 mM NaCl and it's well known that Cas9 would lose its activity when working at high salt concentrations like 2M/4M LiCl.*

The binding of the dCas9 to the target DNA done in physiological low salt conditions as described in the methods section (Line 13, Page 403):

“To assemble the dCas9 RNPs, the tracrRNA (200 nM), crRNA (250 nM) and dCas9 (100 nM) were incubated in a low salt buffer (25 mM HEPES-NaOH (pH 8.0), 150 mM NaCl, 1 mM MgCl₂) at 25 degrees Celsius for at least 20 minutes.”

Once the binding is complete, sample is transferred into the higher salt conditions. Our protocol ensures that we indeed probe the binding activity of dCas9 initially on its target. Figure S5 was added to show the effect of time in the high salt conditions on the dCas9 remaining bound to the DNA, A discussion on the binding efficiency measured in the high salt conditions as compared to the low salt is discussed in the caption.

Figure S5. The DNA nanostructure from Figure 1A was tested with the purple probe (Figure 2A, 3C), also called Probe 1 (Figure 4A) in both 2M and 4M LiCl. It The bound dCas9 is found to be more efficiently remain bound for longer at higher efficiencies in 2M than 4M LiCl. This agrees with results previously shown in the literature.¹⁰

4. Enzymes are sensitive to salt conditions so the initial binding of the dCas9 probe to the target DNA is performed in low salt conditions similar to physiological conditions. The binding efficiency presented in this paper reflects iist's the predicted

binding efficiency when measured in 2M LiCl which is shown in Figure S5 to remain relatively constant over time. This can lead us to believe this the 2M LiCl buffer is not strongly affecting the binding., However in 4M LiCl the binding is found to decrease over time. It can be seen from Figure S5 that at the beginning of the measurement in 4M LiCl the observed binding efficiency is similar to that measured in 2M LiCl, which can also lead us to believe also suggests that this reflects the real binding efficiency under normal buffer conditions. This also agrees with previous discussions in the literature.¹⁰ In Fig. 3A, the highlighted position seems to be 16 nt away from the PAM instead of 14.

This has been fixed.

5. In Fig. 3C, does the GACCACUACCGA above the heatmap indicate the gRNA sequence? How could the target base pair have the highest binding ratio for combinations of dG-rG, dC-rC, dTrU, dA-rA? Shouldn't dG-rC, dC-rG, dT-rA, dA-rU be the highest (almost value of 1)?

This figure has been changed with the heatmap removed to make the experiment more clear. The normalization process was also expanded on.

6. On page 8, the 2nd paragraph says: "For all experiments, $N > 45$ clearly labelled distinguishable events." Does this mean 45 events were enough for analysis? Would this suffer from significant Poisson noise?

Zhu et al perform calculations to determine the maximum number of events needed in the supplementary of their work 'Image Encoding Using Multi-Level DNA Barcodes with Nanopore Readout'. The number of events for every experiment has been added to the figure captions.

Additionally, the following has been added in the Analysis of Nanopore data in the methods section (Line 440, Page 14)::

"Our lab has shown that as few as 4 events are sufficient for positive detection in the majority of cases, while 9 correct events increase the probability of positive detection to more than 90%.³

7. In the caption of Fig. 4, labels like (a), (b), and (c) should be capitalized to keep consistency.

This has been fixed.

8. In Fig. 4A, for probe 2, gRNA is labeled in DNA format, which should be UCAUGUCUUACCCCAAUAA. Same issue for Fig. 4D.

The version that was sent out to the reviewers was an older version than that which was uploaded after the authors were made aware it was being sent out for review. This had been addressed in that version and can also be seen in this version.

9. In Fig. 4F, is the binding ratio data from Fig. 4E? It seems to be not consistent with each other. For mutation in positions 1 to 4, the ratio is 0.2, 0.2, 0.3, 0.8 in Fig. 4E, and 0.2, 0.4, 0.1, 0.8 in Fig. 4E.

This figure has been changed to remove the heatmaps to make it clearer. Additional text was added into the axes to also make it clearer. Figure 4E shows the binding efficiency of the

different probes to their target DNA where the values in Figure 4F are the binding ratios to mismatched DNA. There was a mistake in Figure 4F, which most likely lead to the confusion, that has been fixed:

dG:rC	dT:rC
dG:rC	dT:rC
dC:rG	dA:rG

This: dG:rC Should have been this: dT:rC

The target DNA sequences for each of the probes has also been added now.

10. On page 10, the middle of 1st paragraph says: “Probe 2 showed to have very little ... sensitivity to single base pair mutations ... this lack of specificity highlights that this probe would not be useful as a tool for detecting single base pair changes in clinical samples.” Then why choose Probe 2 for the mutation study in Fig. 4D, E, F?

Probe 2 was selected to see if mutations in the RNA could result in increased specificity. For the study in Figure 4D, four new probes are introduced which are mutated versions of Probe 2. Because probe 2 performed so poorly in terms of specificity, we were interested to see if we slightly altered one base of the probe, would it yield similar results or increase the specificity? It proved to be the latter, highlighting the behaviour depends greatly on the gRNA. As seen when a mutation was introduced in the gRNA at position 1, there was an increased specificity compared to a mutation in the target DNA at position 1. The binding to mismatches changed from ~ 0.7 to ~ 0.2 , showing that probes have guide intrinsic mismatch tolerance and different binding specificity.

11. On page 12, the “Nanopore Fabrication and Measurement” part says: “The measurement is then taken until around 1000 folded and unfolded events are gathered”. How to define the unfolded events, and what do they look like?

Figure S7 was added to the supplementary (Line 108, Page 15) with the following text to further explain.

Figure S7. Unfolded 11011 barcode (A) and Folded 11011 barcode (B) nanopore traces taken from the control measurement in Figure 2B.

Because DNA is non-rigid, it can enter the nanopore folded. One can see that when a DNA event is folded, as it is on the right in Figure S7B, the time scale becomes shorter compared to that of an unfolded event figure S7A. One can also see the initial current drop becomes around double that of the expected DNA baseline of the dsDNA translocating. DNA knotting and folding is commonly observed on solid-state nanopore sensing systems.¹¹

12. On page 13, the end of 1st paragraph says: “Following the sorting, the events are analysed by eye and events which have folds or knots interfering with the barcode are discarded.” Is there any objective way to inspect events instead of manually checking?

The sorting can be further optimized to eliminate folds and knots by applying additional parameters. However, since the data analysis is based on thresholding, this involves creating data-set specific parameters. Because nanopores of different sizes are used, this complicates applying one set of parameters. The amount of time it takes to optimize these parameters for one data set make it much more efficient to do an initial sorting and the last step manually. In the future with more automated data taking the analysis will be automated.

13. In equation (10), what does n mean?

Equation 10 has now been changed to $X_{Control} = \frac{\sum X_{ControlPosition\ 1,2,3}}{3}$ (10) to be more clear.

The 3, previously, n , represents the number of overhangs on the nanostructure. For the control there are 3 overhangs with the same target sequence that are averaged.

References

- 1 Labun, K. *et al.* CHOPCHOP v3: expanding the CRISPR web toolbox beyond genome editing. *Nucleic Acids Research* **47**, W171-W174, doi:10.1093/nar/gkz365 (2019).
- 2 Chen, K. *et al.* Digital Data Storage Using DNA Nanostructures and Solid-State Nanopores. *Nano Letters* **19**, 1210-1215, doi:10.1021/acs.nanolett.8b04715 (2019).
- 3 Zhu, J., Ermann, N., Chen, K. & Keyser, U. F. Image Encoding Using Multi-Level DNA Barcodes with Nanopore Readout. *Small* **17**, 2100711, doi:<https://doi.org/10.1002/sml.202100711> (2021).
- 4 Krysler, A. R., Cromwell, C. R., Tu, T., Jovel, J. & Hubbard, B. P. Guide RNAs containing universal bases enable Cas9/Cas12a recognition of polymorphic sequences. *Nature Communications* **13**, 1617, doi:10.1038/s41467-022-29202-x (2022).
- 5 Pacesa, M. *et al.* Structural basis for Cas9 off-target activity. *Cell* **185**, 4067-4081.e4021, doi:<https://doi.org/10.1016/j.cell.2022.09.026> (2022).
- 6 O'Geen, H., Henry, I. M., Bhakta, M. S., Meckler, J. F. & Segal, D. J. A genome-wide analysis of Cas9 binding specificity using ChIP-seq and targeted sequence capture. *Nucleic Acids Res* **43**, 3389-3404, doi:10.1093/nar/gkv137 (2015).
- 7 Bengtson, M. *et al.* CRISPR-dCas9 based DNA detection scheme for diagnostics in resource-limited settings. *Nanoscale* **14**, 1885-1895, doi:10.1039/D1NR06557B (2022).
- 8 Yi, J.-Y. *et al.* Simple visualization method for the c.577del of erythropoietin variant: CRISPR/dCas9-based single nucleotide polymorphism diagnosis. *Drug Testing and Analysis* **n/a**, doi:<https://doi.org/10.1002/dta.3438>.

- 9 Antony, J. S., Roberts, S. A., Wyrick, J. J. & Hinz, J. M. dCas9 binding inhibits the initiation of base excision repair in vitro. *DNA Repair* **109**, 103257, doi:<https://doi.org/10.1016/j.dnarep.2021.103257> (2022).
- 10 Weckman, N. E. *et al.* Multiplexed DNA identification using site specific dCas9 barcodes and nanopore sensing. *ACS sensors* **4**, 2065-2072 (2019).
- 11 Plesa, C. *et al.* Direct observation of DNA knots using a solid-state nanopore. *Nature Nanotechnology* **11**, 1093-1097, doi:10.1038/nnano.2016.153 (2016).